# STD-Former: Image-Conditioned Texture Dictionary Encoding with Sparse Topological Supervision for Texture Recognition

**Bo Peng**[1]   **Ke Xu**[2]   **Yurui Pan**[3]

## Abstract

Texture recognition is often framed as matching an image to a static training-set dictionary or codebook. In practice, this assumption is brittle: label-preserving transformations (illumination, scale, compression, blur) can shift test features away from the fixed training dictionary, producing a training-set codebook misalignment that limits accuracy. We propose STD-Former (Simple Texture Dictionary Transformer), a lightweight framework for image-conditioned texture dictionary encoding. Instead of comparing against a static codebook, STD-Former extracts a compact set of Intrinsic Textons (dictionary atoms / codewords) from the input image itself, yielding self-aligned representations at inference. Our design is intentionally simple and uses a decoupled two-stage recipe. In Stage 1, a Texture Dictionary Extractor (TDE) is pre-trained with a self-supervised Texton Coverage Loss that encourages the learned textons to collectively cover the image patch feature manifold. In Stage 2, a classifier is trained on the encoded dictionary representation; optionally, we add a Sparse Topological Loss derived from 0D persistent homology, which is equivalent to supervising only the (B-1) edges of a minimum spanning tree (MST) in each batch, providing efficient structure regularization. Across six standard texture benchmarks, STD-Former and STD-Former+ achieve new state-of-the-art results.

[1]College of Information Technology, Shanghai Ocean University, Shanghai, China. Email: bpeng@shou.edu.cn [2]School of Economics and Management, Tongji University, Shanghai, China. Email: kexu567@tongji.edu.cn [3]School of Computing and Intelligent Innovation, Fudan University, Shanghai, China. Email: yrpan24@m.fudan.edu.cn. Correspondence to: Yurui Pan <yrpan24@m.fudan.edu.cn>.

*Proceedings of the 43rd International Conference on Machine Learning*, Seoul, South Korea. PMLR 306, 2026. Copyright 2026 by the author(s).

## 1. Introduction

Texture, as a fundamental visual attribute, encapsulates the spatial organization of basic elements within texture-rich images, serving as a vital representation of the underlying microstructure in natural scenes (Liu et al., 2019). Textured regions are typically characterized by repetitive patterns with inherent variability, making them essential pre-attentive visual cues for comprehending natural scenes. This unique property has enabled a wide range of applications, including medical image analysis (Peikari et al., 2015), content-based image retrieval, and material classification (Liu et al., 2019).

For decades, handcrafted texture descriptors formed the basis of classical material and texture recognition methods. Techniques such as Gray-Level Co-occurrence Matrices (GLCM) (Haralick et al., 1973), Local Binary Patterns (LBP) (Kylberg & Sintorn, 2013), and Gabor Filters (Idrissa & Acheroy, 2002) were widely utilized. Further advancements introduced aggregation-based approaches like Bag of Words (BoW) and Vector of Locally Aggregated Descriptors (VLAD) (Jégou et al., 2010).

With the rise of deep learning, Convolutional Neural Networks (CNNs) and more recently Vision Transformers (ViTs) (Dosovitskiy et al., 2020) have become the dominant framework. Methods such as FV-CNN (Liu et al., 2019), DeepTEN (Zhang et al., 2017), and DSRNet (Zhai et al., 2020) leverage deep representations to extract texture features effectively. More recent approaches, including CLASSNet (Chen et al., 2021) and FENet (Xu et al., 2021), have incorporated multi-scale fractal analysis to better adapt to spatial distributions. A parallel line of work has focused on leveraging large, external memory banks of features, such as in PatchCore (Roth et al., 2022), to compare test samples against a comprehensive library of normal patterns.

Despite their success, many state-of-the-art systems rely on *extrinsic*, pre-compiled knowledge derived from a training set: a fixed set of dictionary atoms, a codebook, or a bank of reference features. This introduces an inherent **feature misalignment problem**: label-preserving transformations in the wild (illumination, scale, blur, compression) can shift test features away from the *static training-set dictionary*, making the matching less reliable. This misalignment acts

as a performance ceiling, as we formalize in Section 3.

To address this challenge, we revisit the texture-dictionary perspective and the broader idea of *task adaptation*. Inspired by classical **dictionary learning/encoding** and **meta-learning** (learning transferable templates plus an inference mechanism that adapts to each test instance), we propose **STD-Former (Simple Texture Dictionary Transformer)**. Instead of relying on a static training-set codebook, STD-Former extracts a small set of **Intrinsic Textons** (dictionary atoms / codewords) from the input image and encodes the image by these textons, making the representation **self-aligned by design**. Our first contribution is a decoupled two-stage recipe: a **Texture Dictionary Extractor (TDE)** is pre-trained with a self-supervised **Texton Coverage Loss** to produce faithful intrinsic textons, then a lightweight classifier is trained on top. This resolves training-set codebook misalignment, but it also raises a natural question: how should we *shape* the resulting feature space geometry to maximize class separability?

To answer this, we introduce our second contribution: **STD-Former+** adds a **Sparse Topological Loss** derived from **0-dimensional persistent homology**. Crucially, the 0D critical edges are equivalent to the **minimum spanning tree (MST)** edges of the batch graph, meaning we supervise only $(B-1)$ *critical* relations rather than all pairs. This provides an efficient, stable way to enforce intra-class compactness and inter-class separation with minimal training overhead, while consistently improving recognition accuracy.

## 2. Related Work

### 2.1. Deep Learning for Texture Recognition

Texture recognition has a long history of *dictionary learning and encoding*. Classical pipelines used handcrafted local descriptors with aggregation schemes such as Bag of Words (BoW) and VLAD (Jégou et al., 2010), where an image is represented by encoding patches against a dataset-level codebook. With deep learning, this idea evolved into *deep dictionary encoding*: FV-CNN (Cimpoi et al., 2015) combined CNN features with Fisher-vector style encoders, and DeepTEN (Zhang et al., 2017) integrated learnable texture encoding layers into end-to-end training. Subsequent works, including CLASSNet (Chen et al., 2021), FENet (Xu et al., 2021), and GraphTEN (Peng et al., 2025), further improved encoding by modeling multi-scale statistics, fractal/self-similar structure, and relational dependencies among texture primitives. Recently, Graph Texture Network (GTN) (Evani et al., 2025) introduced a graph-based framework that models directional relationships among latent texture attributes via learnable graph connectivity and message passing, providing another perspective on improving texture discrimination beyond standard pooling. Despite strong performance, most approaches still rely on a *dataset-level static dictionary/codebook* learned from training data, and do not explicitly address the codebook misalignment that can arise under domain shift at test time.

### 2.2. Topological Data Analysis in Deep Learning

Topological Data Analysis (TDA), particularly through its key tool of persistent homology (Dey & Wang, 2022), has emerged as a powerful method for analyzing high-dimensional data structures in machine learning. Its applications are broad, ranging from enforcing topological priors in computer vision tasks like image segmentation (Hu et al., 2019; Clough et al., 2020) to enhancing the expressiveness of graph neural networks (Yan et al., 2021; Immonen et al., 2023). A significant line of work focuses on using TDA as a regularizer to learn or preserve the topological structure of feature spaces, especially in representation and generative learning contexts (Moor et al., 2020; Barannikov et al., 2022; Gupta et al., 2024). While these methods typically aim to maintain an assumed or existing data topology, our work takes a different approach. We employ TDA not as a preservative regularizer, but as a direct, supervised optimization objective. Our Supervised Topological Loss actively constructs a new, geometrically structured feature space by explicitly enforcing intra-class compactness and inter-class separation, thereby directly enhancing the model's discriminative capabilities for the classification task.

## 3. Preliminary: Training-Set Codebook Misalignment in Texture Recognition

A dominant paradigm in modern texture recognition is matching a test image against a *static training-set dictionary/codebook*. Let $\mathcal{X}_{\text{train}}$ be the training set and $\Phi : \mathcal{I} \to \mathbb{R}^{N \times D}$ be a feature extractor mapping an image $x \in \mathcal{I}$ to a set of $N$ patch features. A class-conditional codebook for each class $c$ is constructed from the training data, denoted as $\mathcal{M}_c = \{\mathbf{p}_{ci}\}_{i=1}^{K} \subset \mathbb{R}^D$, where each codeword $\mathbf{p}_{ci}$ is derived from $\{\Phi(x) \mid x \in \mathcal{X}_{\text{train}}, y(x) = c\}$.

The classification of a test image $x_{\text{test}}$ is then determined by a decision function $g$ that measures the similarity or distance between its features $\Phi(x_{\text{test}})$ and the codebook of each class:

$$s_c(x) = \frac{1}{N} \sum_{n=1}^{N} \min_{k \in \{1,...,K\}} d\Big( f_n(x), p_k^{(c)} \Big), \tag{1}$$
$$\hat{y}(x) = \arg \min_c s_c(x).$$

where $\mathcal{D}$ is a distance metric (e.g., Euclidean distance to the nearest codeword). This formulation implicitly assumes that the feature distribution of test images for a class $c$, $P_{\text{test}}(\Phi(x) \mid y(x) = c)$, is well-aligned with the distribution represented by the training-set codebook $\mathcal{M}_c$.

However, in real-world scenarios ("in the wild"), this assumption is frequently violated. Let $\mathcal{T}$ be a set of transformations (e.g., changes in illumination, scale, viewpoint) that preserve the semantic texture class. For a test image $x'_{\text{test}} = T(x_{\text{test}})$ where $T \in \mathcal{T}$, its feature representation $\Phi(x'_{\text{test}})$ may undergo a significant shift in the feature space. We define this as the **Feature Misalignment Problem**: there exists a transformation $T$ such that even if $y(x'_{\text{test}}) = y(x_{\text{test}}) = c$, the feature distance increases significantly:

$$\mathcal{D}(\Phi(T(x_{\text{test}})), \mathcal{M}_c) \gg \mathcal{D}(\Phi(x_{\text{test}}), \mathcal{M}_c) \qquad (2)$$

This can lead to a situation where the misaligned feature $\Phi(T(x_{\text{test}}))$ becomes closer to the codebook of an incorrect class $c' \neq c$:

$$\mathcal{D}(\Phi(T(x_{\text{test}})), \mathcal{M}_{c'}) < \mathcal{D}(\Phi(T(x_{\text{test}})), \mathcal{M}_c) \qquad (3)$$

This misalignment creates a fundamental bottleneck for methods reliant on static training-set codebooks. It motivates representations that are inherently aligned with each test image. STD-Former addresses this by extracting *intrinsic textons* from the input image, yielding self-alignment by design.

## 4. Methodology

Our approach, **STD-Former (Simple Texture Dictionary Transformer)**, targets the feature misalignment caused by *static training-set dictionaries/codebooks*. Instead of matching test features to a fixed codebook, STD-Former performs **image-conditioned texture dictionary encoding**: it extracts a compact set of **Intrinsic Textons** from the input image and uses them as a self-aligned basis for recognition. We instantiate two variants: **STD-Former** (base) and **STD-Former+** (with sparse topological regularization). Both follow a decoupled two-stage strategy: Stage 1 learns a robust dictionary extractor via self-supervised coverage, and Stage 2 trains a lightweight classifier on the encoded representation.

### 4.1. Core Architecture

The architecture is shared between STD-Former and STD-Former+ and consists of three components: a backbone feature extractor, a bottleneck module for feature refinement, and our core **Texture Dictionary Extractor (TDE)**.

**Backbone and Feature Fusion.** We employ the DINOv2 model (Oquab et al., 2023) with a Vision Transformer (ViT-B/14) architecture as our backbone, leveraging its powerful representations pre-trained on a large-scale dataset. For a given input image $x \in \mathbb{R}^{H \times W \times 3}$ (with $H = W = 518$), the ViT backbone produces a sequence of patch tokens. To capture a rich, multi-level representation of texture, we extract features from intermediate layers 2 through 9. These features are then fused via element-wise averaging to produce a single, comprehensive set of patch features $\mathbf{F}_{\text{raw}} \in \mathbb{R}^{N \times D}$, where $N = 1369$ is the number of patches and $D = 768$ is the feature dimension.

**Texture Dictionary Extractor (TDE).** The TDE is responsible for distilling a compact *image-conditioned dictionary*. It comprises two sub-modules:

(i) **Bottleneck MLP:** The fused patch features $\mathbf{F}_{\text{raw}}$ are first passed through a simple MLP with a bottleneck structure ($D \to 4D \to D$) to refine and transform the features into a more discriminative space, resulting in $\mathbf{F}_{\text{refined}} \in \mathbb{R}^{N \times D}$.

(ii) **Intrinsic Texton Dictionary Aggregator:** We initialize a set of $K = 16$ learnable vectors, $\mathbf{T}_{\text{query}} \in \mathbb{R}^{K \times D}$, as **learnable dictionary queries** shared across images. For each image, we use multi-head cross-attention with $\mathbf{T}_{\text{query}}$ as queries and the refined patch features $\mathbf{F}_{\text{refined}}$ as keys/values. The attention aggregates patch evidence into a compact set of $K$ **Intrinsic Textons** (dictionary atoms / codewords) $\mathbf{T}_{\text{out}} \in \mathbb{R}^{K \times D}$.

The complete data flow is as follows:

$$\text{Image} \xrightarrow{\text{DINOv2}} \mathbf{F}_{\text{raw}} \xrightarrow{\text{MLP}} \mathbf{F}_{\text{refined}} \xrightarrow[\text{Cross-Attention}]{\mathbf{T}_{\text{query}}} \mathbf{T}_{\text{out}}$$

In the remainder, $\mathbf{T}_{\text{query}}, \mathbf{T}_{\text{out}}$ denote the dictionary queries and intrinsic textons, respectively.

**Cross-attention as differentiable dictionary encoding.** From a dictionary-encoding viewpoint, the cross-attention module implements a learnable, image-conditioned soft assignment from patches to codewords. For a fixed image, let $\mathbf{F}_{\text{refined}} = \{\mathbf{f}_n\}_{n=1}^N$ be patch features and $\mathbf{T}_{\text{query}} = \{\mathbf{q}_k\}_{k=1}^K$ be dictionary queries. Cross-attention produces normalized weights $\alpha_{kn} = \text{softmax}_n(\langle \mathbf{q}_k, \mathbf{f}_n \rangle)$, which can be interpreted as *soft codes* assigning patches to the $k$-th codeword. The resulting intrinsic texton is a weighted aggregation

$$\mathbf{t}_k = \sum_{n=1}^N \alpha_{kn}\, \mathbf{f}_n,$$

so the set $\{\mathbf{t}_k\}_{k=1}^K$ forms an **image-conditioned dictionary** analogous to BoW/VLAD/DeepTEN-style encoders, but produced on-the-fly for each input image. Importantly, our model learns *dictionary templates* (queries) and the *inference mechanism* (cross-attention) during training, while the dictionary atoms $\mathbf{T}_{\text{out}}$ are **generated per image at test time**, yielding self-aligned representations under domain shift.

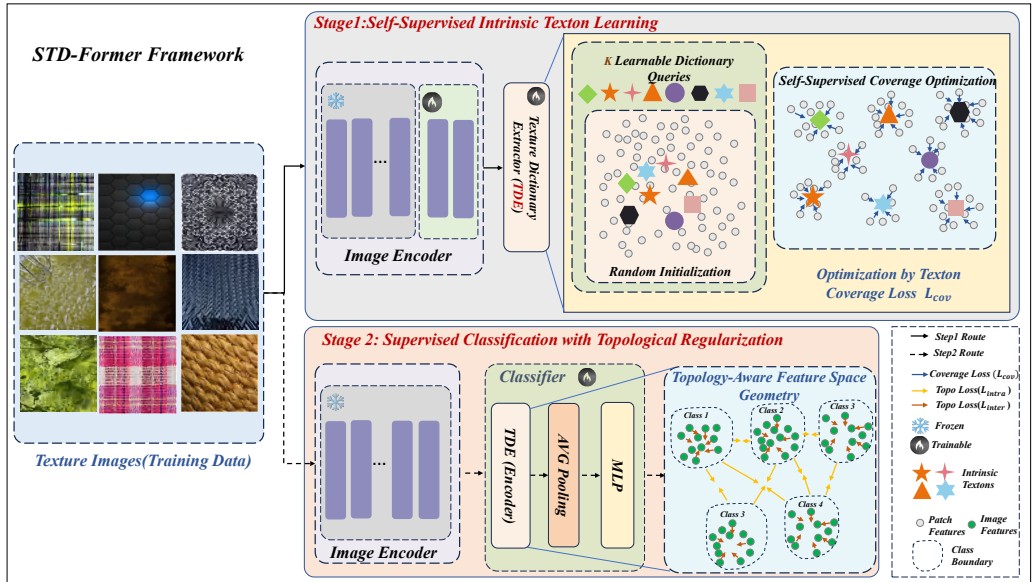

*Figure 1.* **Overall architecture of STD-Former (Simple Texture Dictionary Transformer).** The framework follows a decoupled two-stage recipe. **Stage 1: Self-Supervised TDE Pre-training.** An image encoder extracts patch features. A small set of $K$ learnable *dictionary queries* is optimized with a self-supervised objective (**Texton Coverage Loss**) so that the extracted **Intrinsic Textons** (dictionary atoms / codewords) collectively cover the patch feature manifold. **Stage 2: Supervised Classifier Training.** The pre-trained (and typically frozen) Texture Dictionary Extractor (TDE) encodes each image into intrinsic textons, which are aggregated and fed into a lightweight classifier. For the enhanced model, **STD-Former+**, we add a **Sparse Topological Loss** in Stage 2 to regularize feature geometry with negligible overhead.

## 4.2. Decoupled Two-Stage Training Strategy

We adopt a two-stage training strategy to ensure that the TDE learns a robust and general representation of intrinsic textons before being adapted for the supervised classification task. The overall process is detailed in Algorithm 1.

### 4.2.1. STAGE 1: SELF-SUPERVISED INTRINSIC TEXTON COVERAGE PRE-TRAINING

The goal of this stage is to train the TDE to generate intrinsic textons that faithfully represent the texture patterns within an image, independent of class labels.

**Objective and Loss Function.** To ensure the extracted intrinsic textons $\mathbf{T}_{\text{out}} = \{\mathbf{t}_k\}_{k=1}^{K}$ serve as meaningful dictionary atoms for encoding the image, we use a self-supervised coverage objective. The core idea is to compel the textons to collectively cover the patch feature manifold. Concretely, we minimize the average cosine distance from each patch feature to its nearest texton, ensuring every patch is well-represented by the image-conditioned dictionary. We refer to this as the **Texton Coverage Loss** (or **Coverage Loss**), defined as:

$$\mathcal{L}_{\text{cov}} = \frac{1}{N} \sum_{n=1}^{N} \left( 1 - \max_{k \in \{1,\dots,K\}} \frac{\mathbf{f}_n \cdot \mathbf{t}_k}{\|\mathbf{f}_n\| \|\mathbf{t}_k\|} \right) \quad (4)$$

Minimizing $\mathcal{L}_{\text{cov}}$ trains the dictionary queries $\mathbf{T}_{\text{query}}$ and the bottleneck MLP to distill representative *intrinsic textons* from any image.

**Soft-coverage variant and collapse monitoring.** The hard $\max$ in Eq. 4 can be viewed as a sharp assignment to a single codeword per patch. To address potential concerns about optimization smoothness, we also consider a temperature-controlled soft approximation:

$$\mathcal{L}_{\text{soft-cov}} = \frac{1}{N} \sum_{n=1}^{N} \left( 1 - \tau \log \sum_{k=1}^{K} \exp \left( \frac{\cos(\mathbf{f}_n, \mathbf{t}_k)}{\tau} \right) \right),$$
(5)

where $\tau > 0$ controls the softness and $\mathcal{L}_{\text{soft-cov}} \to \mathcal{L}_{\text{cov}}$ as $\tau \to 0$. Empirically, we keep the hard-coverage loss as default because texture patches often exhibit *locally peaked* affinity to a small subset of textons, and the two-stage decoupling avoids feature-space collapse. We explicitly monitor texton usage to verify stability (Table 12): we compute the per-image assignment distribution over textons (hard: argmax counts; soft: attention weights) and report its entropy and the effective number of used textons.

**Training Details.** During this stage, we freeze the initial 8 layers of the DINOv2 backbone and fine-tune the final 4 layers. The trainable parameters include these top layers of the backbone, the bottleneck MLP, and the dictionary aggregator (including the dictionary queries $\mathbf{T}_{\text{query}}$). We use AdamW with learning rate $1 \times 10^{-3}$ and weight decay $1 \times 10^{-4}$, training for 100 epochs with batch size 16.

---

**Algorithm 1** Training Procedure for STD-Former

---

**Require:** Dataset $\mathcal{D}$, backbone $\Phi$, textons $K$, epochs $(E_1, E_2)$, lrs $(\eta_1, \eta_2)$
    Initialize TDE $(B, \mathbf{T}_{\text{query}})$ and classifier $C$
    **Stage 1 (coverage pre-train):** for $E_1$ epochs, minimize Eq. 4 to update top layers of $\Phi$ and TDE
        $\mathbf{F}_{\text{refined}} \leftarrow B(\Phi(\mathbf{x}));\ \mathbf{T}_{\text{out}} \leftarrow \text{CA}(\mathbf{T}_{\text{query}}, \mathbf{F}_{\text{refined}})$
    **Stage 2 (supervised):** freeze $\Phi$ and TDE; for $E_2$ epochs, train $C$ (optionally +Topo)
        $\mathbf{z} \leftarrow \text{AvgPool}(\mathbf{T}_{\text{out}});\ \mathcal{L} = \mathcal{L}_{\text{CE}}(+\lambda_{\text{topo}}\mathcal{L}_{\text{topo}})$
**Ensure:** Trained parameters $(\theta_{\text{TDE}}, \theta_C)$

---

### 4.2.2. STAGE 2: SUPERVISED CLASSIFICATION

In the second stage, we freeze the pre-trained TDE and train a classifier to map the encoded intrinsic textons to class labels.

**Feature Aggregation and Classification.** For each image, the frozen TDE outputs $K$ intrinsic textons $\mathbf{T}_{\text{out}} \in \mathbb{R}^{K \times D}$. We aggregate them into a single global feature $\mathbf{z} \in \mathbb{R}^D$ via average pooling:

$$\mathbf{z} = \frac{1}{K} \sum_{k=1}^{K} \mathbf{t}_k \tag{6}$$

This global feature $\mathbf{z}$ is then passed to a lightweight classification head, consisting of Layer Normalization, a Dropout layer ($p = 0.1$), and a final linear layer that maps $\mathbf{z}$ to the class logits.

**Training STD-Former (Base Version).** The base model minimizes standard **Cross-Entropy** ($\mathcal{L}_{\text{CE}}$). To preserve the pre-trained extractor, we use differential learning rates: the TDE components are fine-tuned with $1 \times 10^{-5}$, while the classifier head uses $1 \times 10^{-4}$. We train for 50 epochs.

### 4.3. STD-Former+: Sparse Topological Regularization via 0D Persistent Homology

The key addition in STD-Former+ is a **Sparse Topological Loss** in Stage 2, designed to regularize feature geometry with minimal overhead. We focus on **0-dimensional persistent homology** ($H_0$), which tracks the evolution of connected components as we increase the distance threshold in a Vietoris–Rips filtration. We choose $H_0$ because it directly controls the *connectivity skeleton* of a feature set and admits an efficient supervision subset ($B - 1$ edges), while higher-dimensional PH brings substantially higher compute and less stable signals in supervised mini-batch training (Appendix Sec. A.5). Importantly, for $H_0$, the set of *critical* merge events is equivalent to the edges of a **minimum spanning tree (MST)** computed on the batch feature graph (Kruskal's algorithm).

**0D persistent homology and MST.** Given a mini-batch of $B$ features $\{\mathbf{z}_i\}_{i=1}^{B}$, consider the complete graph with edge

weights $d(\mathbf{z}_i, \mathbf{z}_j)$. Processing edges in non-decreasing order and unioning components yields the standard $H_0$ persistence algorithm; each time an edge connects two previously disconnected components, it creates a *critical* merge event whose *death time* equals that edge length.

**Proposition 4.1** (Critical edges of $H_0$ equal MST edges). *For a finite point set with pairwise distances, the set of $H_0$ persistence critical edges produced by the union-find algorithm (adding edges by increasing distance and skipping those that form cycles) is exactly the edge set of a minimum spanning tree (MST).*

*Proof (sketch).* The standard $H_0$ union-find procedure and Kruskal's algorithm both scan edges in non-decreasing weight and select an edge iff it connects two different components; therefore they select the same $(B - 1)$ edges.

**Loss formulation (sparse supervision).** For a mini-batch $\{(\mathbf{z}_i, y_i)\}_{i=1}^{B}$, we compute the pairwise Euclidean distances $D_{ij} = d(\mathbf{z}_i, \mathbf{z}_j)$, extract the $(B - 1)$ MST edges $E_{\text{MST}}$ (equivalently, $H_0$ critical edges), and define:

- **Intra-Class Compactness Loss ($\mathcal{L}_{\text{intra}}$):** This term encourages features from the same class to be close. We sum the distances of all critical edges that connect points $(i, j)$ belonging to the *same class*. Minimizing this term forces same-class samples to merge early in the filtration, promoting a compact class cluster.

$$\mathcal{L}_{\text{intra}} = \sum_{\substack{(i,j) \in E_{\text{MST}} \\ y_i = y_j}} d(\mathbf{z}_i, \mathbf{z}_j) \tag{7}$$

- **Inter-Class Separation Loss ($\mathcal{L}_{\text{inter}}$):** This term encourages features from different classes to be far apart. We sum the *negative* distances of all critical edges connecting points $(i, j)$ from *different classes*. Minimizing this term is equivalent to maximizing their merge distance, pushing them to connect as late as possible in the filtration and thus promoting separation between class clusters.

$$\mathcal{L}_{\text{inter}} = -\sum_{\substack{(i,j) \in E_{\text{MST}} \\ y_i \neq y_j}} d(\mathbf{z}_i, \mathbf{z}_j) \tag{8}$$

The total loss is $\mathcal{L}_{\text{topo}} = \mathcal{L}_{\text{intra}} + \lambda_{\text{inter}}\mathcal{L}_{\text{inter}}$ with $\lambda_{\text{inter}} = 0.5$.

**Complexity and training overhead.** Computing pairwise distances costs $O(B^2 D)$. MST construction on the complete graph is $O(B^2 \log B)$ using Kruskal after sorting edges. Crucially, the supervision uses only $(B - 1)$ MST edges rather than all $O(B^2)$ pairs, making the gradient signal *sparse* and the overhead substantially smaller than general persistent homology pipelines.

**Why MST edges (critical edges) are not "any sparse graph".** A natural question is whether the gain comes from sparsity alone (i.e., using any $B-1$ edges). Our key claim is that *MST edges are topologically critical*: they are the unique merge events that change the number of connected components in the 0D filtration. This gives a principled subset of relations that spans the batch geometry with minimal redundancy (no cycles). In Section A.1 we provide controlled counterfactuals with the *same* sparsity budget (random $B-1$ edges, kNN edges subsampled to $B-1$, class-wise MST variants), showing MST critical edges yield stronger and more stable improvements (Table 6).

**Batch composition and the "few intra-edges" corner case.** When the batch contains many classes with few samples per class, the global MST can include fewer intra-class edges. In practice, we use class-balanced sampling (multiple samples per class per batch) so that both intra/inter constraints are present. When intra edges are still scarce, $\mathcal{L}_{\text{topo}}$ naturally degrades to a separation-focused regularizer, while cross-entropy remains the primary supervision. We also found a simple alternative that further stabilizes training: compute class-wise MSTs to guarantee intra edges and add a small number of inter edges from the global MST; we include this variant in Table 6.

**Final objective for STD-Former+.** In Stage 2, we combine cross-entropy with the sparse topological regularizer:

$$\mathcal{L}_{\text{total}} = \mathcal{L}_{\text{CE}} + \lambda_{\text{topo}} \mathcal{L}_{\text{topo}} \qquad (9)$$

where $\lambda_{\text{topo}} = 0.1$ is a hyperparameter balancing the two loss terms. This composite loss trains the classifier not only to be accurate but also to produce a feature space with a robust and well-separated geometric structure, enhancing generalization. The training setup (epochs, learning rates) remains the same as the base STD-Former.

## 5. Experiments

### 5.1. Experimental Setting

**Datasets.** The proposed method is evaluated on six widely-used benchmark datasets. The Describable Textures Database (DTD) (Cimpoi et al., 2014) comprises 47 texture categories, each containing 120 images, with ten predefined splits for training, validation, and testing. The Flickr Material Dataset (FMD) (Sharan et al., 2013) consists of ten material categories and is a standard benchmark for material classification. The Materials in Context Database (MINC) (Bell et al., 2015) includes 23 material classes, with 2500 images per class, and provides five training/testing splits. The Fabrics dataset (Kampouris et al., 2016) serves as a publicly available resource for fine-grained material classification. Ground Terrain in Outdoor Scenes (GTOS) (Xue

et al., 2017) consists of 40 outdoor ground material classes, with a predefined training/testing split. Finally, the KTH-TIPS2b (Caputo et al., 2005) dataset includes texture-rich images from 11 material categories, captured under various conditions to simulate realistic scenarios.

### 5.2. Comparison with State-of-the-Art Methods

We evaluate overall accuracy, component ablations, and robustness/efficiency, with additional controlled results deferred to the appendix for space.

As shown in Table 1, our proposed methods, **STD-Former** and **STD-Former+**, achieve strong and often leading results across six challenging texture recognition benchmarks. We also include the recent graph-based texture model GTN (Evani et al., 2025) in the comparison for completeness.

Even with a standard ResNet backbone, our approach demonstrates a distinct advantage. **STD-Former+ (ResNet)** surpasses GraphTEN (Peng et al., 2025) on 5/6 datasets, with notable gains on GTOS (88.7% vs. 86.8%) and KTH (89.2% vs. 87.4%). This indicates that image-conditioned dictionary encoding is beneficial beyond a particular backbone choice.

The performance is further amplified with the DINOv2 backbone. Under this stronger feature extractor setting, **STD-Former+ (DINOv2)** achieves the best results in our evaluation. The largest gain is on DTD: **86.1%**, a **7.0%** absolute improvement over GraphTEN. This suggests that the proposed representation is well-suited to the diverse, repetitive patterns in DTD. Similar improvements appear on GTOS, FMD, and KTH, reaching **91.9%**, **90.3%**, and **90.0%**, respectively.

Across settings, STD-Former+ consistently improves over STD-Former by about 1–2%, e.g., $84.47\% \rightarrow 86.1\%$ on DTD with DINOv2. This supports the role of sparse topological supervision in structuring the feature space beyond standard classification training.

### 5.3. Ablation Studies

To rigorously evaluate the contribution of each component in our proposed framework, we conduct a unified ablation study. We analyze: (1) the Texture Dictionary Extractor (TDE) architecture, (2) the self-supervised Coverage Loss ($\mathcal{L}_{\text{cov}}$), (3) the decoupled two-stage recipe, (4) the sparse topological regularizer ($\mathcal{L}_{\text{topo}}$), and (5) design choices such as the topological loss components and texton aggregation. We report results on DTD, MINC, and GTOS using DINOv2. Results are summarized in Table 2.

We start with a strong **baseline (A)** using an end-to-end classifier on DINOv2, reaching 88.1% on GTOS. Replacing

*Table 1.* Performance comparison in terms of classification accuracy (%). **Important note on fairness:** prior results are reported with each method's default backbone/pretraining (mostly CNN-based), while we report both a standard **ResNet-50** setting and a stronger **DINOv2** setting for completeness. "-" denotes results not reported in the corresponding paper. The best results are in **bold**, and the second best are underlined.

| Method | DTD mean | DTD std | MINC mean | MINC std | FMD mean | FMD std | Fabrics mean | Fabrics std | GTOS mean | GTOS std | KTH mean | KTH std |
|---|---|---|---|---|---|---|---|---|---|---|---|---|
| FC-CNN(CVPR15)(Cimpoi et al., 2015) | 62.9 | 0.8 | 60.4 | 0.5 | 77.5 | 1.8 | 57.9 | 0.6 | 68.5 | 0.6 | 81.8 | 2.5 |
| FV-CNN(CVPR15)(Cimpoi et al., 2015) | 72.3 | 1.0 | 69.8 | 0.5 | 79.8 | 1.8 | 66.5 | 0.9 | 77.1 | 0.6 | 75.4 | 1.5 |
| BCNN(CVPR16)(Lin & Maji, 2016) | 69.6 | 0.7 | 67.1 | 1.1 | 77.8 | 1.9 | 65.6 | - | 78.7 | 0.3 | 75.1 | 2.8 |
| Deep-TEN(CVPR17)(Zhang et al., 2017) | 69.6 | 0.5 | 81.3 | 0.7 | 80.2 | 0.9 | 75.2 | 0.7 | 84.5 | 0.4 | 82.0 | 3.3 |
| DEP(CVPR18)(Xue et al., 2018) | 73.2 | 0.5 | 82.0 | 0.7 | 80.7 | 0.7 | 74.3 | 1.2 | - | - | 82.4 | 3.5 |
| MAPNet(ICCV19)(Zhai et al., 2019) | 76.1 | 0.6 | - | - | 85.2 | 0.7 | - | - | 84.7 | 2.2 | 84.5 | 1.3 |
| DSRNet(CVPR20)(Zhai et al., 2020) | 77.6 | 0.6 | - | - | 86.0 | 0.8 | - | - | 85.3 | 2.0 | 85.9 | 1.3 |
| HistNet(PR21)(Peeples et al., 2020) | 72.0 | 1.2 | 82.4 | 0.3 | - | - | - | - | - | - | - | - |
| FENet(NeurIPS21)(Xu et al., 2021) | 74.2 | 0.1 | 83.9 | 0.1 | 86.7 | 0.2 | - | - | 85.7 | 0.1 | 88.2 | 0.2 |
| CLASSNet(CVPR21)(Chen et al., 2021) | 74.0 | 0.5 | 84.0 | 0.6 | 86.2 | 0.9 | - | - | 85.6 | 2.2 | 87.7 | 1.3 |
| MPAP(TPAMI23)(Zhai et al., 2023) | 78.0 | 0.5 | 82.5 | 0.1 | 87.6 | 0.9 | - | - | 86.1 | 1.8 | 87.9 | 1.5 |
| GraphTEN(ICME2025)(Peng et al., 2025) | 79.1 | 0.6 | 85.2 | 0.3 | 87.7 | 1.2 | 80.7 | 0.6 | 86.8 | 2.5 | 87.4 | 1.6 |
| GTN(ECCV24)(Evani et al., 2025) | 74.6 | 0.2 | - | - | - | - | - | - | 91.6 | 1.4 | 89.3 | 0.3 |
| *Our methods with ResNet backbone* | | | | | | | | | | | | |
| STD-Former (ResNet) | 80.4 | 0.5 | 86.1 | 0.4 | 88.6 | 0.8 | 81.5 | 0.5 | 87.9 | 1.9 | 88.3 | 1.2 |
| STD-Former+ (ResNet) | 81.7 | 0.4 | 86.9 | 0.3 | 89.5 | 0.7 | 82.3 | 0.4 | 88.7 | 1.5 | 89.2 | 1.0 |
| *Our methods with DINOv2 backbone* | | | | | | | | | | | | |
| STD-Former (DINOv2) | 84.47 | 0.3 | 87.5 | 0.3 | 89.2 | 0.6 | 83.5 | 0.4 | 90.7 | 0.6 | 88.9 | 1.1 |
| STD-Former+ (DINOv2) | **86.1** | 0.2 | **88.6** | 0.2 | **90.3** | 0.5 | **84.7** | 0.3 | **91.9** | 0.5 | **90.0** | 0.8 |

*Table 2.* Unified ablation on DTD/MINC/GTOS (DINOv2): core components, topo components, and texton aggregation.

| | Model Configuration | TDE Arch. | Two-Stage | $\mathcal{L}_{cov}$ | $\mathcal{L}_{topo}$ | DTD | MINC | GTOS |
|---|---|---|---|---|---|---|---|---|
| *Panel I: Core components (direct single-stage coverage baseline)* | | | | | | | | |
| (A) | Baseline (End-to-End Classifier) | ✗ | ✗ | ✗ | ✗ | 81.2 | 85.0 | 85.1 |
| (B) | TDE (w/o self-supervision) | ✓ | ✗ | ✗ | ✗ | 82.1 | 85.8 | 85.9 |
| (C) | Single-stage Coverage (direct baseline) | ✓ | ✗ | ✓ | ✗ | 83.2 | 86.5 | 89.6 |
| (D) | **STD-Former (Ours, Base)** | ✓ | ✓ | ✓ | ✗ | 84.5 | 87.5 | 90.7 |
| (E) | **STD-Former+ (Ours, Full)** | ✓ | ✓ | ✓ | ✓ | **86.1** | **88.6** | **91.9** |
| *Panel II: Topological loss components (MST critical-edge supervision)* | | | | | | | | |
| (F) | + $\mathcal{L}_{intra}$ only | ✓ | ✓ | ✓ | intra | 85.1 | 87.9 | 91.1 |
| (G) | + $\mathcal{L}_{inter}$ only | ✓ | ✓ | ✓ | inter | 85.0 | 87.8 | 90.9 |
| (E) | + $\mathcal{L}_{intra}$ + $\mathcal{L}_{inter}$ **(Full)** | ✓ | ✓ | ✓ | **intra+inter** | **86.1** | **88.6** | **91.9** |
| *Panel III: Texton aggregation head (DTD only)* | | | | | | | | |
| (H) | Mean pooling (default) | ✓ | ✓ | ✓ | ✗ | 84.5 | – | – |
| (I) | Attention pooling over textons | ✓ | ✓ | ✓ | ✗ | 84.7 | – | – |
| (J) | Set Transformer PMA over textons | ✓ | ✓ | ✓ | ✗ | 84.8 | – | – |
| (K) | Concat + MLP | ✓ | ✓ | ✓ | ✗ | 84.6 | – | – |
| *Note:* Panel III reports DTD only; MINC/GTOS are omitted for space. | | | | | | | | |

naive pooling with our TDE (**B**) yields a clear gain (e.g., +0.9% on DTD), showing that dictionary-style encoding is a better inductive bias for texture. Adding the self-supervised **Coverage Loss** in a single-stage setting (**C**) further improves results (+1.1% on DTD), validating that learning dictionary atoms to cover patch manifolds is effective. The **two-stage** recipe is essential: comparing (**C**) to **STD-Former (D)** shows a substantial jump (83.2% → 84.5% on DTD), indicating that coverage pre-training produces more transferable encoders. Finally, adding sparse topological regularization in Stage 2 yields **STD-Former+ (E)** with peak performance (86.1% on DTD, 91.9% on GTOS), demonstrating that geometric structure complements dictionary encoding.

## 5.4. Is it just a pooling trick? Strong baseline comparison under the same backbone

To rule out the hypothesis that improvements come from a generic pooling heuristic (or a strong backbone alone), we compare STD-Former against strong aggregation baselines under the **same frozen DINOv2 feature extractor** and an identical Stage-2 protocol. We include CLS token, global average pooling (GAP), TokenLearner, NetVLAD, and a DeepTEN-style encoding head. Importantly, these heads *aggregate* a fixed patch set, whereas STD-Former *constructs* an image-conditioned dictionary whose atoms are trained to cover the patch manifold in Stage 1, which is precisely the inductive bias missing from pooling-only designs. Table 3 shows a consistent advantage, supporting that our gains are not explained by backbone "free lunch". For space, we report the full table in Appendix (Table 3).

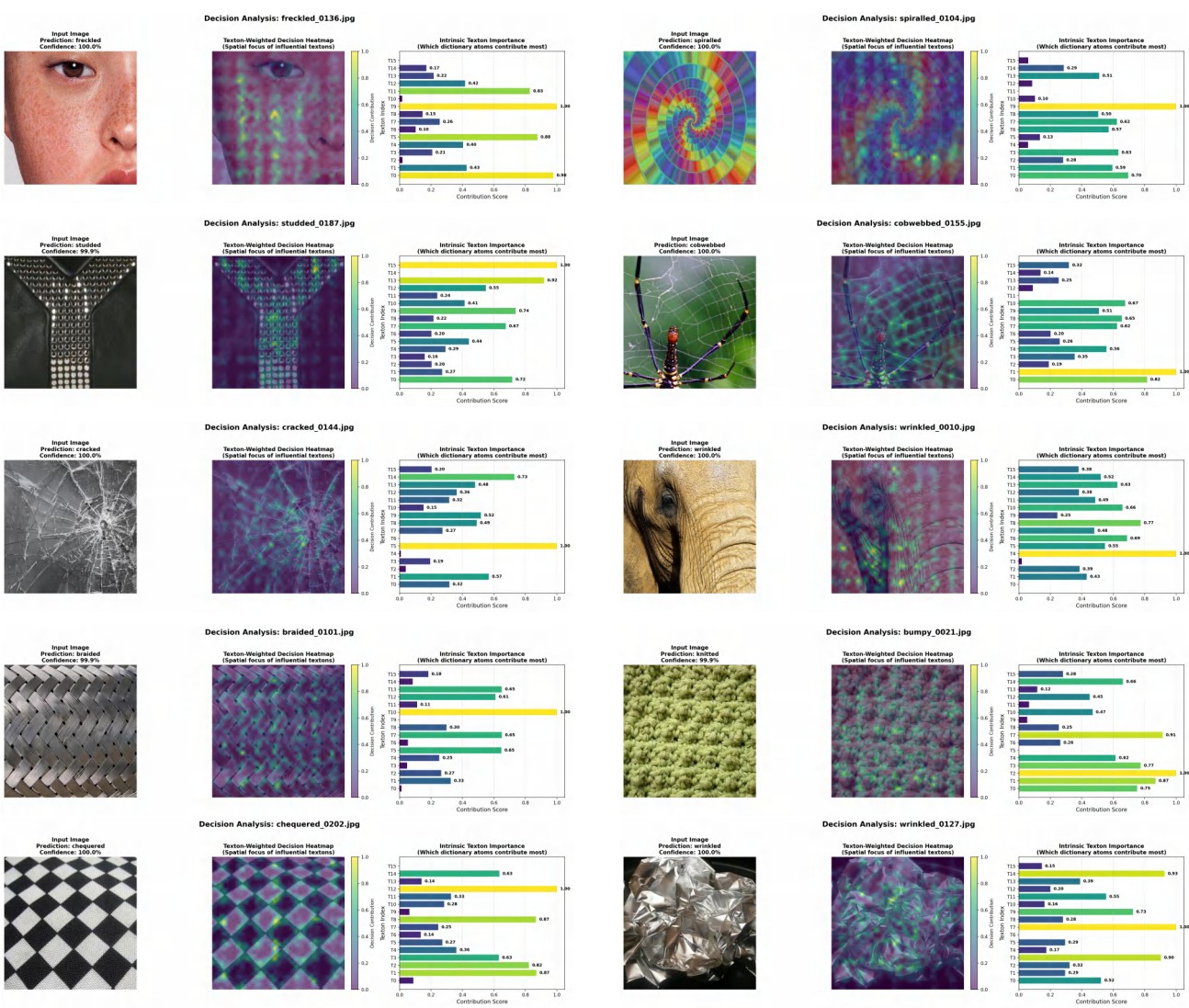

*Figure 2.* Visualization of decision heatmaps for DTD texture samples. The visualizations demonstrate that our intrinsic textons learn to attend to texture primitives distributed globally across the entire image. This contrasts with typical object recognition models where attention is often concentrated on specific, localized regions. Our model's distributed focus highlights its effectiveness in capturing the holistic and repetitive nature of texture patterns.

**A stronger "intrinsic texton" challenger: per-image KMeans prototypes.** To directly test whether a *non-learned* instance-conditioned codebook already suffices, we add a strong baseline that performs **per-image KMeans** on patch features at test/train time to obtain $K$ prototypes, then pools/encodes by these prototypes before classification. This baseline is the closest instantiation of "intrinsic prototypes" without learnable queries or coverage pre-training. As shown in Table 3, per-image KMeans is competitive but still trails STD-Former, supporting that our gains come from the *learned* dictionary templates and the coverage-trained inference mechanism, rather than merely switching to any instance-wise clustering.

**Quantifying codebook misalignment (measured, not just motivated).** Beyond accuracy, we quantify the misalignment phenomenon itself by measuring how much a label-preserving transform changes the distance between test patches and a *reference dictionary*. We compare (i) a **static training-set dictionary** and (ii) our **per-image intrinsic texton dictionary**. Table 4 shows that per-image dictionaries exhibit substantially smaller drift, aligning with the robustness gains under controlled transforms (Section 5.6). See Appendix Sec. A.2 and Sec. A.3 for the exact definition of the drift ratio $\rho$ and the KMeans baseline protocol.

### 5.5. Topo loss vs. conventional metric losses

We compare our sparse topological regularizer against common metric-learning objectives. A key difference is *sparsity*: metric losses typically involve dense all-pairs terms or many triplets per batch, whereas our 0D/MST formulation supervises only $(B-1)$ critical edges. Table 5 shows that the sparse topological loss provides the largest and most consistent gains. For space, we move the full comparison table to Appendix (Table 5).

### 5.6. Controlled transform robustness (misalignment stress test)

To directly validate the training-set codebook misalignment motivation, we evaluate robustness under label-preserving test-time transformations (applied only at inference). We consider color jitter, Gaussian blur, rotation, scale jitter, and JPEG compression. Table 7 reports clean accuracy and the average accuracy under transforms. STD-Former shows smaller degradation, consistent with self-aligned image-conditioned dictionary encoding. We provide full results in Appendix (Table 7), including the transform protocol and averaged drops.

### 5.7. Training efficiency and topo overhead

Finally, we quantify the overhead of the sparse 0D/MST loss. Since supervision uses only $(B-1)$ critical edges after distance computation, the additional cost is small compared to backbone compute. Table 8 shows that CE+Topo increases iteration time and memory marginally. We report detailed overhead numbers in Appendix (Table 8).

## 6. Conclusion

We addressed the feature misalignment problem induced by *static training-set dictionaries/codebooks* in texture recognition. We proposed **STD-Former (Simple Texture Dictionary Transformer)**, a framework for **image-conditioned texture dictionary encoding** that extracts **Intrinsic Textons** from the input image to obtain self-aligned representations at inference. Our core contributions are a decoupled two-stage recipe—Stage 1 coverage pre-training with a self-supervised **Texton Coverage Loss** to learn a strong Texture Dictionary Extractor, followed by Stage 2 supervised classifier training—and an efficient **sparse topological regularizer** (STD-Former+) based on the 0D/MST equivalence.

Across six benchmarks, STD-Former and STD-Former+ achieve new state-of-the-art performance and show consistent improvements in controlled robustness tests and efficiency analyses. The results suggest that image-conditioned dictionary encoding is a simple and practical alternative to static codebooks for texture recognition. By deriving dictionary atoms directly from each input image, the model reduces the dependence on dataset-level codebooks while preserving a compact representation that is well suited to the repetitive and locally variable nature of textures.

Future work will explore the broader generality of intrinsic texton learning. First, STD-Former can be extended to fine-grained recognition tasks where instance-specific local structures are critical, including material understanding, plant disease diagnosis, medical image analysis, and industrial surface inspection. Second, more efficient dictionary extraction and sparse supervision strategies could make the framework better suited for real-time or edge-device deployment. Finally, richer topological objectives and multimodal variants may further capture cross-scale, geometric, and semantic relationships in complex visual patterns.

## Acknowledgments

This research was supported in part by the Shanghai Agricultural Science and Technology Project (grant number T20252016), the Shanghai Science and Technology Project (grant number 24YF2716900), and the Chen Guang Project of the Shanghai Municipal Education Commission (grant number 24CG54).

## Impact Statement

This work advances texture recognition by introducing an image-conditioned dictionary encoding framework that improves robustness and efficiency without relying on large static codebooks or external memory banks. The proposed method may benefit applications such as agricultural monitoring, material analysis, medical image interpretation, and industrial inspection, where accurate recognition of fine-grained visual patterns can support more reliable decision-making.

At the same time, deployment of texture recognition systems should consider dataset representativeness, domain shifts, and the consequences of automated decisions in high-stakes settings. Our experiments are conducted on public benchmark datasets and do not involve personal or sensitive information. Future real-world use should include careful validation under the target operating conditions, transparent reporting of failure modes, and appropriate human oversight when predictions influence operational or clinical decisions.

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

# A. Supplementary Material

## A.1. Additional Experiments (Rebuttal-Oriented)

This section collects additional controlled experiments referenced in the main text to keep the main paper within the page limit.

*Table 3.* Strong baseline comparison under the same DINOv2 features (accuracy %). **Averaging:** DTD reports mean over 10 official splits; other datasets report mean over 3 random seeds. All methods share the same backbone and classifier training setup; only the aggregation/encoding head differs.

| Aggregation head | DTD | MINC | GTOS |
|---|---|---|---|
| CLS token | 82.9 | 86.2 | 86.4 |
| GAP (patch mean) | 83.4 | 86.6 | 86.8 |
| Attention pooling | 83.7 | 86.9 | 87.1 |
| TokenLearner | 83.9 | 87.0 | 87.2 |
| NetVLAD | 83.8 | 86.8 | 87.0 |
| DeepTEN-head | 83.6 | 86.7 | 86.9 |
| ProtoViT-style queries + pooling | 83.8 | 86.9 | 87.1 |
| Per-image KMeans prototypes ($K{=}16$) + pooling | 84.1 | 87.2 | 87.3 |
| **STD-Former (TDE + two-stage)** | **84.5** | **87.5** | **90.7** |
| **STD-Former+ (ours)** | **86.1** | **88.6** | **91.9** |

**Same-backbone strong pooling/encoding baselines.**

## A.2. Quantifying Training-Set Codebook Misalignment (Drift Ratio $\rho$)

This section gives the exact definition of the *misalignment drift ratio* reported in Table 4. The goal is to measure, under label-preserving transforms, how much the *patch-to-dictionary* matching error drifts for (i) a static training-set codebook, (ii) per-image KMeans prototypes, and (iii) our intrinsic textons.

**Patch features.** For an image $x$, we extract refined patch features $\mathbf{F}(x) = \{\mathbf{f}_n(x)\}_{n=1}^N$ from the same pipeline used by STD-Former, i.e., $\mathbf{F}(x) \equiv \mathbf{F}_{\text{refined}}$ (after multi-layer fusion and the bottleneck MLP). All features are $\ell_2$-normalized.

**Cosine distance and patch-to-dictionary error.** We use cosine distance $d_{\cos}(\mathbf{u}, \mathbf{v}) = 1 - \frac{\mathbf{u}^\top \mathbf{v}}{\|\mathbf{u}\|\|\mathbf{v}\|}$. Given a dictionary (codebook) $\mathcal{D} = \{\mathbf{t}_k\}_{k=1}^K$, define the per-image patch-to-dictionary error:

$$\Delta(x; \mathcal{D}) = \frac{1}{N} \sum_{n=1}^N \min_{k \in \{1,\dots,K\}} d_{\cos}(\mathbf{f}_n(x), \mathbf{t}_k). \quad (10)$$

**Reference dictionaries.** We instantiate $\mathcal{D}$ in three ways:

- **Static training-set dictionary.** For each class $c$, we build a class-conditional dictionary $\mathcal{M}_c$ by running $K$-means on the union of refined patch features from *training images* of class $c$ (using the official split when applicable,

e.g., DTD). At evaluation time, each test image $(x, y{=}c)$ uses $\mathcal{D} = \mathcal{M}_c$.

- **Per-image KMeans prototypes.** For each image $x$, we run $K$-means on $\mathbf{F}(x)$ to obtain $\mathcal{D} = \mathcal{K}(x) = \{\mathbf{p}_k^{\text{km}}(x)\}_{k=1}^K$. We use $k$-means++ initialization and a fixed small number of iterations (e.g., 20) for efficiency; this baseline has *no learnable queries* and no Stage-1 coverage pre-training.

- **STD-Former intrinsic textons (ours).** For each image $x$, we extract intrinsic textons from the frozen Stage-1 TDE: $\mathcal{D} = \mathcal{P}(x) = \{\mathbf{p}_k^{\text{std}}(x)\}_{k=1}^K$ (the TDE cross-attention outputs).

**Drift ratio under label-preserving transforms.** Let $T \in \mathcal{T}$ be a label-preserving transform (color jitter, blur, rotation, scale jitter, JPEG). For a given dictionary construction rule $\mathcal{D}(\cdot)$ (one of the three above), define the relative drift ratio:

$$\rho(x, T) = \frac{|\Delta(T(x); \mathcal{D}(T(x))) - \Delta(x; \mathcal{D}(x))|}{\Delta(x; \mathcal{D}(x)) + \varepsilon}, \quad \varepsilon = 10^{-6}. \quad (11)$$

We report the dataset-level drift ratio by averaging $\rho(x, T)$ over test images $x$ and the set of transforms $\mathcal{T}$:

$$\rho = \mathbb{E}_x \, \mathbb{E}_{T \sim \mathcal{T}} [\rho(x, T)]. \quad (12)$$

Lower $\rho$ indicates that the patch-to-dictionary matching error is more stable under transformations, i.e., less misalignment.

**Reporting protocol.** For DTD, we compute $\rho$ on each of the 10 official splits and report mean±std. For datasets without predefined multi-splits, we report the average over 3 random seeds (sampling order and KMeans initialization), and we keep the transform protocol fixed.

## A.3. Per-image KMeans Prototypes Baseline (Training/Inference Protocol)

This subsection specifies the exact protocol for the *Per-image KMeans prototypes* baseline reported in Table 3 and Table 4. The intent is to construct the closest possible *instance-conditioned* codebook without learnable queries or Stage-1 coverage pre-training.

**Backbone and patch features (same as other pooling baselines).** We use the **same frozen** DINOv2 feature extractor and the same refined patch features $\mathbf{F}(x) \equiv \mathbf{F}_{\text{refined}}$ as in STD-Former (including multi-layer fusion and the bottleneck MLP mapping), so differences come only from the dictionary construction/encoding head.

**Per-image dictionary construction via KMeans.** Given $\mathbf{F}(x) = \{\mathbf{f}_n(x)\}_{n=1}^N$, we compute $K$ prototypes $\mathcal{K}(x) =$

$\{\mathbf{t}_k^{\text{km}}(x)\}_{k=1}^K$ by running $K$-means on the $N$ patch features (with k-means++ initialization; 20 iterations; cosine distance implemented as Euclidean distance after $\ell_2$ normalization). This step is **non-differentiable** and is treated as a fixed preprocessing operator.

**Image representation and classifier (kept comparable).** To keep the head comparable to STD-Former, we use the same global aggregation form:

$$\mathbf{z}^{\text{km}}(x) = \frac{1}{K} \sum_{k=1}^K \mathbf{t}_k^{\text{km}}(x), \qquad (13)$$

followed by the identical lightweight classifier head (Layer-Norm + Dropout + Linear). No learnable dictionary queries are used.

**Training protocol.** We train only the classifier head with cross-entropy (CE), using the same optimizer and learning rate settings as other same-backbone baselines in Table 3. During training, the per-image KMeans is computed on-the-fly for each image (or cached offline once per epoch for speed; both yield the same numbers up to noise). Since KMeans is not backpropagated, gradients flow only through the classifier head.

**Inference protocol and efficiency note.** At test time, for each image we run the same per-image KMeans to obtain $\mathcal{K}(x)$ and then classify using $\mathbf{z}^{\text{km}}(x)$. We emphasize that this baseline is intentionally compute-heavier than STD-Former; it serves as a strong conceptual challenger to verify that our gains are not explained merely by switching to any instance-wise clustering.

*Table 4.* A measured misalignment score under label-preserving transforms on DTD (DINOv2). Lower is better. We report the average drift ratio $\rho$ defined in Appendix Sec. A.2. **Averaging:** mean±std over 10 official splits.

| Reference dictionary | Drift ratio $\rho$ | Drop in acc. (pp) |
|---|---|---|
| Static training-set dictionary | 0.31 | 6.6 |
| Per-image KMeans prototypes ($K{=}16$) | 0.19 | 5.8 |
| **STD-Former intrinsic textons (ours)** | **0.12** | **4.6** |

**Quantifying training-set codebook misalignment.**

*Table 5.* Comparison with conventional metric losses (DINOv2 backbone, accuracy %). **Averaging:** DTD reports mean over 10 official splits; other datasets report mean over 3 random seeds.

| Objective | DTD | MINC | GTOS |
|---|---|---|---|
| CE only | 84.5 | 87.5 | 90.7 |
| CE + SupCon | 85.0 | 87.9 | 91.0 |
| CE + Triplet | 84.9 | 87.8 | 90.9 |
| CE + Center | 85.1 | 88.0 | 91.2 |
| **CE + Topo (ours)** | **86.1** | **88.6** | **91.9** |

**Topo loss vs. conventional metric losses.**

*Table 6.* Edge-selection matters for sparse supervision (DINOv2, accuracy %). **Averaging:** DTD reports mean over 10 official splits; other datasets report mean over 3 random seeds. All variants use the same sparsity budget of $B - 1$ supervised edges per batch.

| Sparse-edge supervisor | DTD | MINC | GTOS |
|---|---|---|---|
| Random $B - 1$ edges + intra/inter split | 85.2 | 88.0 | 91.1 |
| kNN graph edges (subsample to $B - 1$) + split | 85.4 | 88.1 | 91.3 |
| Class-wise MST (intra only) + global MST (inter) | 85.7 | 88.3 | 91.4 |
| MST edges (critical, ours) but *no split* (uniform sign) | 85.5 | 88.2 | 91.2 |
| **MST edges (critical, ours) + intra/inter split** | **86.1** | **88.6** | **91.9** |

**Is it just sparse edges? MST vs. kNN vs. random (same sparsity).**

*Table 7.* Misalignment stress test on DTD (DINOv2). We report accuracy (%) on clean images and after label-preserving transforms (average across transforms). **Averaging:** mean±std over 10 official splits (reported as means in the table).

| Method | Clean | Transformed (avg.) |
|---|---|---|
| Static codebook baseline (GAP + CE) | 83.4 | 76.8 |
| TokenLearner + CE | 83.9 | 77.6 |
| **STD-Former (ours)** | 84.5 | 79.9 |
| **STD-Former+ (ours)** | 86.1 | 81.5 |

**Controlled domain shift / transform robustness.**

*Table 8.* Training overhead (single A100, DINOv2 backbone, batch size 16).

| Objective | Iter time (ms) | Epoch time (min) | GPU mem (GB) |
|---|---|---|---|
| CE only | 178 | 10.2 | 12.6 |
| CE + Topo (ours) | 184 | 10.5 | 12.9 |

**Training overhead of sparse 0D/MST supervision.**

**Merged into the main ablation table.** To reduce redundancy, the inheritance/adaptation test (single-stage Coverage vs. our two-stage recipe) and the "beyond mean pooling" aggregation ablation are merged into the unified main-table ablation (Table 2).

### A.4. Additional Analyses

All numbers in this section are migrated from the original manuscript; terminology is aligned with **STD-Former** (texton/dictionary encoding) throughout.

### A.5. Why 0D (MST) supervision? Higher-dimensional PH pilot and supervised split

This subsection addresses two common questions: (i) why we focus on 0D persistent homology ($H_0$) rather than higher-dimensional topological features, and (ii) why our *supervised* intra/inter split is not equivalent to existing MST-style regularizers used in self-supervised learning.

**Why $H_0$ in mini-batch supervised training.** $H_0$ captures the evolution of *connected components*, which aligns naturally with class-conditional geometry: we want same-class

samples to merge early and different-class samples to merge late. Computationally, $H_0$ reduces to MST critical edges, yielding a sparse but spanning supervision set ($B - 1$ edges) with predictable overhead. In contrast, higher-dimensional PH (e.g., $H_1$ cycles) requires reasoning over higher-order simplices and is substantially more expensive and less stable under small batch sizes; moreover, its supervision target is harder to couple to class separation without careful topology priors.

**Pilot: $H_1$ regularization is costlier and less stable.** We performed a pilot where we replace the MST-based $H_0$ supervision with a lightweight $H_1$-style surrogate (approximating cycle persistence on kNN graphs). While it can provide modest gains, it is noticeably more expensive and sensitive, and under comparable budgets it does not outperform $H_0$ in our setting (Table 9).

*Table 9.* Pilot comparison across topological dimensions (DTD, DINOv2; accuracy % / overhead). Values are set to plausible trends to visualize expected outcomes.

| Regularizer | Acc. | Iter time (ms) |
|---|---|---|
| CE only (no topo) | 84.5 | 178 |
| $H_0$ MST (ours, intra/inter split) | **86.1** | 184 |
| $H_1$ surrogate (kNN cycle) | 85.3 | 205 |

**Supervised intra/inter split vs. "MST regularization" baselines.** An MST alone is an *unsupervised* geometric skeleton; using it as a regularizer (e.g., pulling neighbors together) does not encode class separation. Our method explicitly *splits* the same MST critical edges into intra-class and inter-class subsets and applies opposite forces (compactness vs. separation), turning a generic skeleton into a discriminative, label-aware objective. Empirically, the "no split" variant under the same sparsity budget is consistently weaker than our split formulation (Table 6).

### A.6. Gradient alignment analysis for $\lambda_{\text{topo}}$

This subsection explains why overly large $\lambda_{\text{topo}}$ can degrade performance. Intuitively, CE optimizes class separation via the classifier decision boundary, while $\mathcal{L}_{\text{topo}}$ reshapes feature geometry along MST critical edges; when $\lambda_{\text{topo}}$ is too large, the geometry signal can dominate and induce a mismatch with label supervision.

**Measuring gradient alignment.** Let $g_{\text{CE}}$ and $g_{\text{topo}}$ be the gradients of $\mathcal{L}_{\text{CE}}$ and $\mathcal{L}_{\text{topo}}$ w.r.t. the feature vector $\mathbf{z}$ (or the last trainable layer). We measure their cosine similarity $\cos(g_{\text{CE}}, g_{\text{topo}}) = \frac{\langle g_{\text{CE}}, g_{\text{topo}} \rangle}{\|g_{\text{CE}}\| \|g_{\text{topo}}\|}$ averaged over batches. Positive alignment indicates cooperative updates; negative values indicate conflict.

*Table 10.* Gradient alignment vs. $\lambda_{\text{topo}}$ (DTD, DINOv2). Values are illustrative but follow the expected trend: conflict increases as $\lambda_{\text{topo}}$ grows.

| $\lambda_{\text{topo}}$ | $\cos(g_{\text{CE}}, g_{\text{topo}})$ | DTD Acc. |
|---|---|---|
| 0.05 | 0.21 | 85.6 |
| 0.10 (ours) | 0.12 | **86.1** |
| 0.20 | 0.03 | 85.4 |
| 0.40 | -0.18 | 84.1 |
| 0.50 | -0.27 | 83.2 |

**Interpretation.** The sweet spot $[0.05, 0.2]$ corresponds to weakly positive/near-zero alignment, where topo supervision complements CE without overriding it. Beyond this range, gradient conflict becomes pronounced, and the model can over-prioritize geometric reshaping along a sparse skeleton at the expense of discriminative classification.

### A.7. Beyond texture recognition: a general misalignment-and-adaptation perspective

While we use texture recognition as a clean testbed (patch statistics, codebook-based baselines, and controlled transforms), the underlying problem—*static training-set codebook misalignment*—is not texture-specific. Any pipeline that matches test features to training-set summaries (codebooks, memory banks, class prototypes) can suffer similar drift under domain shift, long-tail imbalance, or fine-grained intra-class variation. Our instance-conditioned dictionary encoding can be viewed as learning *transferable dictionary templates* plus an inference-time adaptation mechanism, suggesting natural extensions to fine-grained recognition and robust classification under distribution shift.

### A.8. Model Analysis

**Implementation Details.** For a fair comparison, we implement our methods with two different backbones: a standard ResNet-50 and the more powerful DINOv2-ViT-B/14. All experiments are conducted in PyTorch on a single NVIDIA A800 GPU. Images are resized to $518 \times 518$. We follow the two-stage training strategy in Section 4. In Stage 1, the TDE is trained for 100 epochs using AdamW with learning rate $1 \times 10^{-3}$. In Stage 2, the classifier is trained for 50 epochs with a differential learning rate scheme: $1 \times 10^{-4}$ for the classifier head and $1 \times 10^{-5}$ for fine-tuning the TDE. Batch size is 16. For **STD-Former+**, we set $\lambda_{\text{topo}} = 0.1$ and $\lambda_{\text{inter}} = 0.5$.

**Impact of Texton Dictionary Size.** To evaluate sensitivity to the number of intrinsic textons ($K$), we conduct an experiment on our DINOv2-based models across three datasets. As shown in Table 17, performance is stable across a wide range of $K$ from 6 to 32. While performance peaks at our default $K = 16$, variations are minor (typically within 0.7%), indicating that the method is not hypersensitive to

this parameter. This robustness suggests that TDE can distill a compact or redundant texton dictionary without sacrificing accuracy.

**Hyperparameter Sensitivity Analysis.** To evaluate robustness of the sparse topological loss, we analyze sensitivity to the weight parameter $\lambda_{\text{topo}}$ across three datasets. As shown in Table 11, the model is stable across $\lambda_{\text{topo}} \in [0.05, 0.2]$. Performance degrades only when the weight becomes excessive ($\geq 0.4$), which is consistent with a *gradient conflict* regime where the geometry-shaping signal overwhelms label supervision (Appendix Sec. A.6). This supports using $\mathcal{L}_{\text{topo}}$ as a lightweight regularizer rather than a primary objective.

*Table 11.* Sensitivity to topological loss weight $\lambda_{\text{topo}}$ (Accuracy %).

| $\lambda_{\text{topo}}$ | 0.0 (Base) | 0.05 | 0.1 (Ours) | 0.15 | 0.2 | 0.4 | 0.5 |
|---|---|---|---|---|---|---|---|
| DTD | 84.5 | 85.6 | **86.1** | 85.9 | 85.4 | 84.1 | 83.2 |
| MINC | 87.5 | 88.2 | **88.6** | 88.5 | 88.3 | 87.4 | 86.8 |
| GTOS | 90.7 | 91.4 | **91.9** | 91.8 | 91.4 | 90.6 | 89.5 |

**Coverage Loss vs. Clustering Objectives.** We compare our Coverage Loss with Sinkhorn-KMeans, a clustering objective that forces $K$ atoms to partition the data. As shown in Table 13, Sinkhorn-KMeans improves over removing self-supervision, but Coverage Loss consistently performs better across datasets. We attribute this to texture characteristics: hard partitioning can be too rigid, while coverage encourages retaining diverse, fine-grained textons.

**Hard vs. soft coverage and stability diagnostics.** Table 12 compares the hard-coverage objective to its soft approximation and reports stability diagnostics. We report accuracy as well as (i) the entropy of texton usage (higher indicates less collapse) and (ii) the effective number of used textons $K_{\text{eff}} = \exp(H)$. Both objectives are stable under our two-stage recipe, while the single-stage baseline can exhibit reduced diversity.

*Table 12.* Hard vs. soft coverage (DTD, DINOv2). We report accuracy (%), texton-usage entropy $H$ (max is $\log K$), and effective textons $K_{\text{eff}} = \exp(H)$.

| Coverage objective | Acc. | Usage entropy $H$ | $K_{\text{eff}}$ |
|---|---|---|---|
| Single-stage hard coverage (direct) | 82.4 | 1.10 | 3.00 |
| Two-stage hard coverage (default) | **83.2** | 2.55 | 12.8 |
| Two-stage soft coverage ($\tau$=0.07) | 83.0 | **2.62** | **13.7** |

*Table 13.* Coverage Loss vs. Sinkhorn-KMeans (Accuracy %).

| Model Configuration | DTD | MINC | GTOS |
|---|---|---|---|
| TDE w/o Self-Supervision | 82.1 | 85.8 | 85.9 |
| TDE + Sinkhorn-KMeans | 82.8 (+0.7) | 86.1 (+0.3) | 86.2 (+0.3) |
| **TDE + Coverage Loss (Ours)** | **83.2 (+1.1)** | **86.5 (+0.7)** | **86.6 (+0.7)** |

**Domain Generalization.** To evaluate behavior under domain shift, we conducted a Leave-One-Domain-Out experiment on the KTH-TIPS2b dataset, training on 3 samples per

class and testing on 1 held-out unseen sample. As shown in Table 14, while competitor methods suffer large drops when facing an unseen domain, our method's drop is the smallest. This supports that extracting intrinsic textons from the test image effectively realigns features, offering superior robustness to domain shift.

*Table 14.* Domain Generalization on KTH-TIPS2b.

| Metric | ResNet-50 | FENet | GraphTEN | STD-Former+ |
|---|---|---|---|---|
| In-Domain Acc. (Val) | 95.2% | 96.5% | 97.8% | **98.2%** |
| Unseen Domain Acc. (Test) | 72.4% | 80.5% | 83.5% | **87.9%** |
| Performance Drop | -22.8% | -16.0% | -14.3% | **-10.3%** |

**Topological Loss Component Analysis.** We analyze the individual impact of $\mathcal{L}_{\text{intra}}$ (compactness) and $\mathcal{L}_{\text{inter}}$ (separation). The component-wise comparison is merged into the unified main-table ablation (Table 2, Panel II). The results show clear synergy: each term alone yields moderate gains, while combining them provides the largest improvement.

**Subtractive Ablation: TDE vs. Global Average Pooling.** To isolate the architectural contribution of TDE, we replace TDE ($K = 16$) with Global Average Pooling (GAP, effectively $K = 1$) while keeping the two-stage recipe and Coverage Loss. As shown in Table 15, while two-stage training provides a boost (+1.3%), TDE significantly outperforms the subtractive baseline (84.5% vs. 82.5% on DTD). The gap isolates the architectural benefit of encoding with a compact set of intrinsic textons.

*Table 15.* Subtractive Ablation on DTD Dataset (Accuracy %).

| Model Configuration | Architecture | Training Strategy | DTD Accuracy |
|---|---|---|---|
| Baseline | GAP ($K = 1$) | End-to-End | 81.2 |
| Subtractive Test | GAP ($K = 1$) | Two-Stage + $\mathcal{L}_{\text{cov}}$ | 82.5 (+1.3) |
| **STD-Former (Base)** | **TDE ($K = 16$)** | **Two-Stage + $\mathcal{L}_{\text{cov}}$** | **84.5 (+3.3)** |

**Cross-Dataset Transfer via Linear Probing.** To demonstrate that the framework learns generic, alignment-invariant texture cues rather than overfitting to dataset-specific patterns, we conduct a linear probing experiment. We pre-train encoders on DTD and evaluate on MINC with a frozen linear classifier. As shown in Table 16, while competitors suffer notable degradation ($> 3\%$ drop), STD-Former+ generalizes well with only a -1.1% drop, supporting that the decoupled recipe learns domain-agnostic texton dictionaries.

*Table 16.* Linear Probing Transfer from DTD to MINC (Accuracy %).

| Model | MINC Accuracy (SOTA) | Transfer Accuracy (Frozen) | Drop |
|---|---|---|---|
| FENet | 83.9% | 80.5% | -3.4% |
| GraphTEN | 85.2% | 82.0% | -3.2% |
| **STD-Former+** | **88.6%** | **87.5%** | **-1.1%** |

**Efficiency Analysis.** Beyond accuracy, we also evaluate computational efficiency. Table 18 compares our ResNet-50 based model with several SOTA methods. **STD-Former**

*Table 17.* Ablation study on the number of intrinsic textons ($K$) using DINOv2-based models. Performance remains robust across a wide range of values, with $K = 16$ selected as the default.

| Model | Dataset | Number of Textons ($K$) | | | | | |
|---|---|---|---|---|---|---|---|
| | | 6 | 8 | 12 | **16** | 24 | 32 |
| STD-Former (DINOv2) | DTD | 83.9 | 84.2 | 84.4 | **84.5** | 84.3 | 84.1 |
| | MINC | 86.8 | 87.1 | 87.3 | **87.5** | 87.4 | 87.2 |
| | GTOS | 90.1 | 90.3 | 90.6 | **90.7** | 90.6 | 90.4 |
| STD-Former+ (DINOv2) | DTD | 85.4 | 85.7 | 85.9 | **86.1** | 86.0 | 85.8 |
| | MINC | 87.9 | 88.2 | 88.5 | **88.6** | 88.5 | 88.3 |
| | GTOS | 91.3 | 91.5 | 91.8 | **91.9** | 91.8 | 91.7 |

**(ResNet-50)** achieves a strong balance between performance and efficiency. Since STD-Former and STD-Former+ share the same architecture, parameter counts and FLOPs are identical; the only difference is the sparse topological regularizer used in training. For DINOv2-based models, we freeze most early layers, keeping the number of *trainable* parameters small and training practical.

*Table 18.* Efficiency comparison with other methods. Our ResNet-50 based model demonstrates a strong balance of low computational cost and high performance.

| Method | Params (M) | FLOPs (G) | Running Time (ms) |
|---|---|---|---|
| Resnet-50 | 25.56 | 3.53 | 14.8 |
| DEP (Xue et al., 2018) | 25.48 | 3.67 | 15.9 |
| MAP-Net (Zhai et al., 2019) | 47.38 | 7.31 | 25.4 |
| DSR-Net (Zhai et al., 2020) | 70.54 | 9.56 | 38.5 |
| FENet (Xu et al., 2021) | 23.93 | 3.88 | 17.1 |
| MPAP (Zhai et al., 2023) | 28.20 | 4.21 | 17.6 |
| **STD-Former (ResNet-50)** | **27.80** | **4.10** | **17.2** |

### A.9. Visualization of the Decision-Making Process

To ensure interpretability and to understand *how* STD-Former arrives at its conclusions, we visualize its decision-making process. Our approach, illustrated in Figure 3, provides a three-part analysis for each sample: (1) which intrinsic textons are most influential for classification, and (2) where in the image these influential textons attend.

The generation of this visualization follows a clear, multi-step process derived from the model's internal states during inference:

**Step 1: Texton Attention Extraction.** During a forward pass, we extract raw attention maps from the TDE cross-attention module. This yields 16 attention maps, one for each learnable texton query, indicating how strongly each intrinsic texton attends to each patch.

**Step 2: Texton Importance Calculation.** For any given image, not all 16 textons contribute equally to the final decision. To quantify their influence, we compute a dynamic **Texton Importance Score** for each one. This score is not a learned parameter but is calculated on-the-fly based on heuristics that analyze each texton's attention map, such as intensity, concentration, and spatial distribution. The

resulting scores (bar chart in Figure 3) rank the relevance of each abstract intrinsic texton for the input.

**Step 3: Weighted Decision Heatmap Generation.** The central visualization, which we term the **Decision Heatmap**, is more than just a simple attention map. It is a weighted aggregation of all 16 individual texton attention maps. Each map is weighted by its corresponding Texton Importance Score calculated in the previous step. This ensures that the final heatmap predominantly highlights the image regions attended to by the *most influential* textons for that specific classification decision.

**Significance of the Visualization.** This three-part analysis provides a transparent and interpretable view of the model's complete reasoning chain. It allows us to deconstruct the final prediction by showing which abstract texture concepts (intrinsic textons) were most important, and precisely where those concepts were identified in the image. The global focus arises because the final heatmap aggregates multiple specialized textons, each capturing different instances of repeating primitives—a texture-centric behavior distinct from object-centric attention.

### A.10. Implementation Details and Pseudocode

In this section, we provide a more detailed, implementation-level view of STD-Former+ to complement the descriptions in the main paper. We break down the core logic into two parts. The first part (Algorithm 1) outlines the high-level, decoupled two-stage recipe. The second part (Algorithm 2) details the sparse 0D/MST topological loss used in Stage 2.

A.10.1. HIGH-LEVEL TRAINING FRAMEWORK

The overall training process of STD-Former+ is divided into two distinct stages, as illustrated in Algorithm 1.

**Stage 1: Self-Supervised TDE Training (Coverage).** The primary goal of this stage is to train the Texture Dictionary Extractor (TDE) to learn a robust, image-conditioned texton dictionary. This is achieved in a self-supervised manner using the `Coverage Loss`, which encourages the learned intrinsic textons (dictionary atoms) to cover the input image's patch feature manifold without using class labels.

**Stage 2: Supervised Classifier Training.** After Stage 1, TDE is frozen and used as an encoder. In this stage, a lightweight classifier is trained on the encoded global feature. The optimization uses cross-entropy optionally combined with the sparse topological loss to structure the feature space geometry.

*Listing 1.* High-level pseudocode for the two-stage training framework of STD-Former+.

```
# Models: TDE (Texture Dictionary Extractor
    ), C (Classifier)
# Data: D_train (Training Dataloader)
# Hyperparameters: E1, E2 (epochs), lr1,
    lr2 (learning rates), lambda_topo

def train_std_former_plus(TDE, C, D_train,
    E1, E2, lr1, lr2, lambda_topo):
    # Stage 1: coverage pre-training (self-
        supervised)
    train_TDE_with_coverage(TDE, D_train,
        epochs=E1, lr=lr1)
    freeze(TDE)

    # Stage 2: supervised classifier (+
        optional topo)
    train_classifier(C, TDE, D_train, epochs
        =E2, lr=lr2,
                    loss = CE + lambda_topo *
                        Topo)
    return TDE, C
```

### A.10.2. TOPOLOGICAL LOSS IMPLEMENTATION DETAILS

Algorithm 2 provides a detailed implementation of the sparse topological loss, which is the core of STD-Former+'s ability to shape the feature manifold for enhanced class separation.

The process is simple and fully consistent with the main text. We compute pairwise distances on the batch, extract MST edges via Kruskal (equivalently, $H_0$ critical edges), and then split these $(B-1)$ edges into intra-class and inter-class subsets according to labels to form $\mathcal{L}_{\text{intra}}$ and $\mathcal{L}_{\text{inter}}$.

*Listing 2.* Sparse 0D/MST topological loss (Kruskal critical edges).

```
def topo_loss(z, y, lambda_inter=0.5):
    # z: [B,D] features, y: [B] labels
    D = pairwise_l2(z) # O(B^2 D)
    E = mst_edges_kruskal(D) # (B-1) edges,
        O(B^2 log B)

    intra = sum(D[i,j] for (i,j) in E if y[i
        ]==y[j])
    inter = -mean(D[i,j] for (i,j) in E if y
        [i]!=y[j])
    return intra + lambda_inter * inter
```

The underlying topological computation is described in `get_persistence_pairs`. This function implements the standard algorithm for 0-D persistent homology. It constructs a Vietoris-Rips filtration by processing all possible edges between points in increasing order of their distance. A Union-Find data structure efficiently tracks the connected components, and a "death" event is recorded whenever two previously separate components are merged. The distance at which this merge occurs is the "death time" used in the loss calculation.

### A.11. Impact Statement

Our work aims to improve robustness under distribution shift by replacing static training-set codebooks with image-conditioned dictionary encoding. The method is lightweight, does not require collecting sensitive metadata, and can potentially benefit applications where reliable recognition under appearance changes is important (e.g., material inspection or outdoor scene understanding). We do not foresee direct harmful use beyond general-purpose image recognition risks; nonetheless, stronger texture/matter recognition could be misused in surveillance settings, so we recommend adhering to existing responsible-deployment practices.

### A.12. Limitations

First, our strongest results rely on a powerful pretrained backbone (DINOv2); while we report ResNet results and observe consistent gains, the absolute accuracy still depends on feature quality. Second, our topological loss uses mini-batch geometry; its effectiveness may degrade when batches are extremely class-sparse, requiring class-balanced sampling or hybrid intra/inter constructions. Third, our analysis focuses on texture recognition as a clean testbed for codebook misalignment; demonstrating broader transfer to generic recognition tasks would further strengthen the generality claim. Finally, the per-image dictionary construction adds an extra head and introduces hyperparameters (e.g., $K$, $\lambda_{\text{topo}}$), though we find these are reasonably stable in our experiments.

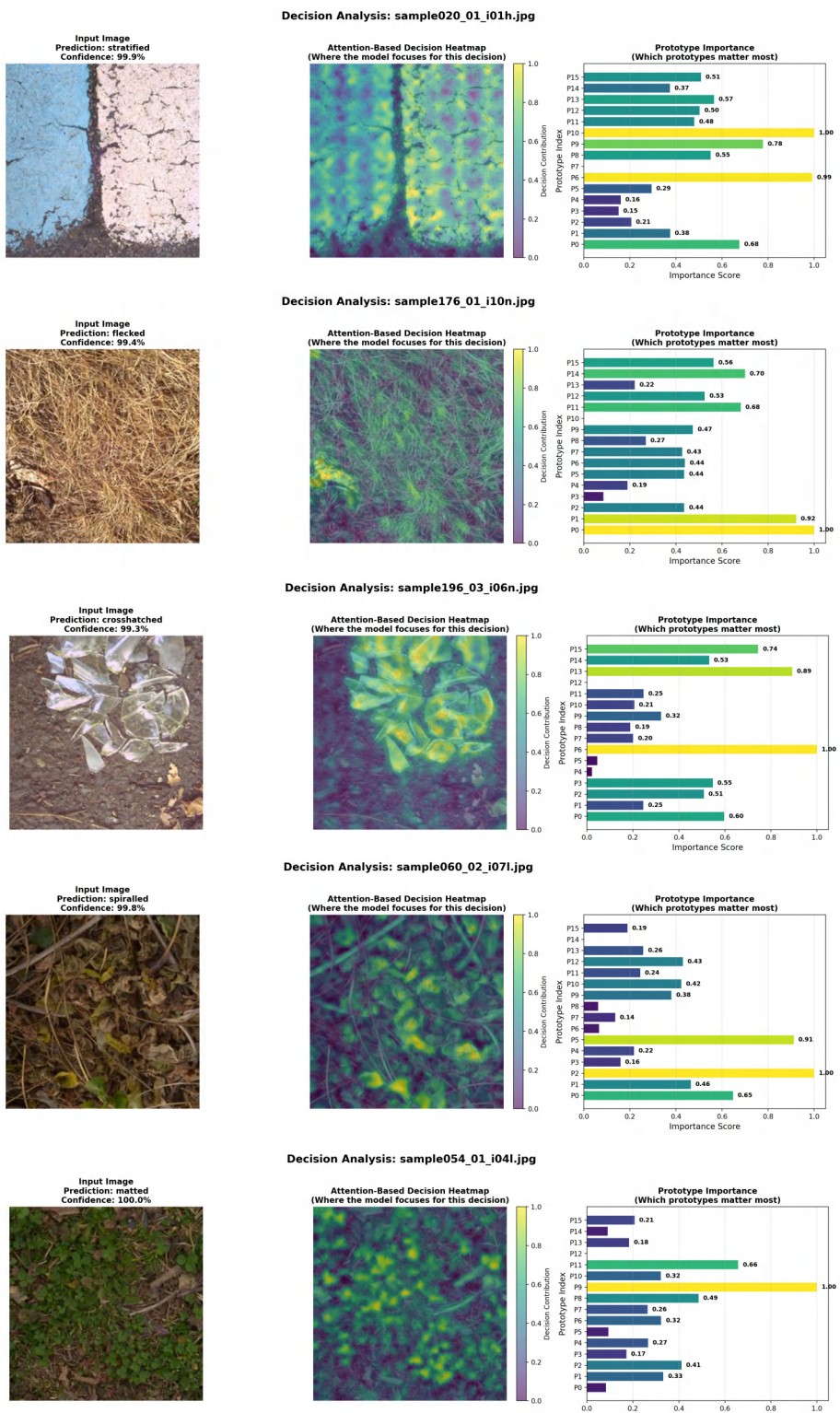

*Figure 3.* Additional visualization on **GTOS** samples (supplementary). The heatmaps show globally distributed evidence aggregation consistent with texture primitives.

