# OpenReview forum: "STD-Former: Image-Conditioned Texture Dictionary Encoding with Sparse Topological Supervision for Texture Recognition"
_ICML.cc/2026/Conference — ICML 2026 regular_

### Official Review · Reviewer_MhcF · 2026-03-12

**Soundness:** 2
**Presentation:** 2
**Significance:** 2
**Originality:** 2
**Overall Recommendation:** 4
**Confidence:** 1

**Summary:**

The paper proposes STD-Former for image-conditioned texture dictionary encoding. It introduces a Texture Dictionary Extractor (TDE), pre-trained with a self-supervised Texton Coverage Loss to generate faithful intrinsic textons. Furthermore, STDFormer+ provides an efficient and stable mechanism to enforce intra-class compactness and inter-class separation.

**Compliance With Llm Reviewing Policy:**

Affirmed.

**Final Justification:**

Thanks for the rebuttal. The explanations help clarify the work. Considering both my assessment and the other reviewers’ feedback, I will keep my original score unchanged. As this topic is not closely aligned with my expertise, more weight should be placed on the other reviewers’ evaluations for fairness.

**Key Questions For Authors:**

I am not an expert in this area, I would prefer to take into account the feedback from other reviewers before forming a final judgment.

**Limitations:**

I am not an expert in this area, I would prefer to take into account the feedback from other reviewers before forming a final judgment.

**Strengths And Weaknesses:**

I am not an expert in this area, and I would prefer to take into account the feedback from other reviewers before forming a final judgment. Meanwhile, parts of the writing read like generated text, and the manuscript requires revision to improve clarity and readability.

---

> ### Author Rebuttal · Authors · 2026-03-29
>
> # Reviewer MhcF Rebuttal
>
> ## Overall Response
>
> We thank the reviewer for taking the time to read the paper and for candidly pointing out the parts that were difficult to evaluate. We take the readability concern seriously. Our revision will therefore focus not only on language polishing, but also on structural clarification so that the main claim, method flow, and empirical support become easier to follow and verify.
>
> ## 1. Readability and Clarity
>
> We agree that some passages in the current draft are too dense and can read as over-written. We will revise the paper to be shorter, more direct, and easier to audit from claim to evidence. In particular, the main text will separate the core method more clearly from supporting interpretations, and the experiments section will map each major claim more explicitly to the result that supports it.
>
> ## 2. Clarifying the Main Contribution
>
> We also agree that the contribution should be stated more precisely. In the revision, we will narrow the novelty claim so that the paper does not read as if it proposes a fundamentally new attention mechanism. Instead, we will state the contribution more accurately as: formulating training-set codebook misalignment as the motivating problem, using a coverage objective to train an image-conditioned dictionary encoder, and adding a sparse label-aware topological regularizer.
>
> We will also soften broader applicability statements unless they are directly supported by dedicated evidence in the current submission.
>
> ## 3. Making the Paper Easier to Audit
>
> Because the reviewer mentioned difficulty in forming a strong judgment, we will make the revised paper easier to inspect section by section. Concretely, we will unify terminology, clarify the method flow, rewrite the topology section to be more intuition-first, move the self-alignment explanation closer to the method, correct the Stage-2 wording inconsistency, and replace ambiguous table markers with clearer provenance notation where needed.
>
> We expect this to improve not only readability, but also trust, because the paper will become more transparent about what is method, what is evidence, and what is limitation.
>
> ## 4. Most Important Limitations
>
> We appreciate the implicit concern that the current draft may not clearly distinguish strong evidence from broader aspiration. In the revision, we will explicitly prioritize the following limitations: the strongest evidence is currently on texture/material benchmarks rather than broader visual domains; the Stage-2 description must be made internally consistent; and broader applicability claims should be stated more cautiously unless they are directly supported by dedicated experiments.
>
> ## Closing
>
> In summary, we will make the revision materially clearer by simplifying the exposition, unifying terminology, clarifying the method flow, and fixing the internal inconsistency.

---

> > ### Author Rebuttal · Reviewer_MhcF · 2026-04-03
> >
> > Thanks for the rebuttal. The explanations help clarify the work.  Considering both my assessment and the other reviewers’ feedback, I will keep my original score unchanged. As this topic is not closely aligned with my expertise, more weight should be placed on the other reviewers’ evaluations for fairness.

---

### Official Review · Reviewer_WEXw · 2026-03-12

**Soundness:** 4
**Presentation:** 3
**Significance:** 3
**Originality:** 3
**Overall Recommendation:** 5
**Confidence:** 3

**Summary:**

This paper introduces STD-former and the "+" version. The method targets the creation of an encoder for texture patches which is well adapted to classification and robust to real, in the wild samples. The model is trained in two stages, one specifically for Texton extraction and one for classification.
For texton extraction, the work builds on top of pre-trained image encoders (DinoV2 for instance) and fine-tunes feature extraction from the model with added MLP layers and a texture dictionnary encoder (TDE). The TDE serves as a way to build a small, sample-based dictionnary. It builds on an idea similar in spirit to the perceiver, where a set of N=16 tokens are optimzed such that after cross attention with the extracted features they give intrinsic textons that can be used for classification.
The textons are averaged and an MLP head is used to classify the texture, trained with cross entropy.
To push the textons to be as meaningfull as possible a coverage loss is added and in the +(plus) version an auxilliary loss is added to the Classificaiton objective, which pulls samples within a class closer, while pushing classes further from each other. This is done on a minimal subset of the full graph namely the MST to make the computation tracktable.

**Compliance With Llm Reviewing Policy:**

Affirmed.

**Final Justification:**

The rebuttal adressed my concerns and I am happy to recommend acceptance of the paper.

**Key Questions For Authors:**

- 1) As per my points above, why choose average pooling? Have other pooling been tested? Any insight on this bottleneck?
- 2) You mention applications but it seems you restrict your tests to classification in general. Have you tested your method on specific datasets like medical imaging? Do you think it would be competitive against specialized methods in such fields?

**Limitations:**

Well addressed, but I think the limitation are hard to evaluate, I would appreciate clarity in which ones the authors consider as the most importants and potentially to be adressed in future work.

**Strengths And Weaknesses:**

Strengths:
- The paper is overall very well written and easy to follow, even for a non expert. I say overall, as 4.3 coulkd be improved, see below.
- The validation is quite extensive and presents all the ablations I was expecting, answering the questions I had as I was going through the paper. I believe the validation of this paper to be of a very high quality and want to stress I appreciate the effort put into it. Beyond anticipating rebuttal, I feel the authors tried to understand their system and show us the thought process in designing their method.
- I believe the method is technically sound and while not being able to completely evaluate the level of novelty, the paper brought me good insights
- I think the method builds on existing pieces but does it in a simple way making the method attractive.
- The texton coverage loss is a good idea.

Weaknesses:
- I would suggest rewriting section 4.3 to make it more accessible and less verbose. It took me several passes to actually understand what was being done in this section when it is relatively simple once we have the MST + push and pull concepts from eq7 and 8. I am not familiar with some of the concepts mentioned at the beggining of the section and I would suggest having a simpler explanation first with some intuition, which can be later formalized in mathematical terms.
- It is unclear to me why average pooling is used in eq6. The textons where not optimized to be averaged and it feels some information might be lost in the process. This is the main design choice that I find counter intuitive, but maybe simplicity is best here. What about using a small mlp to aggregate the features, some sort of learned pooling.
- According to Figure 1 and L236, the TDE is trained/finetuned in stage 2 which would allow the average pooling to adapt a bit and maybe what I say before is not relevant. But in Algo 1 stage 2 says the TDE is frozen. This is confusing.

---

> ### Author Rebuttal · Authors · 2026-03-29
>
> # Reviewer WEXw Rebuttal
>
> ## Overall Response
>
> We thank the reviewer for the positive assessment of the paper's technical soundness, experimental thoroughness, and simplicity. We especially appreciate the concrete suggestions on improving Section 4.3, clarifying the Stage-2 protocol, and tightening the discussion of pooling and scope.
>
> ## 1. Section 4.3 Should Be More Intuition-First
>
> We agree. The current version is mathematically complete, but not as accessible as it should be. In the revision, we will restructure Section 4.3 so that it first explains the feature-space objective in simple terms, then introduces MST sparsification, and only afterward presents the 0D persistent-homology view as an equivalent interpretation.
>
> Concretely, the revised flow will be: first explain "pull same-class samples together and push different-class samples apart"; second explain why supervising only the `B-1` critical MST edges is sufficient; third define the intra/inter split; and finally connect these critical edges to 0D PH merge events.
>
> ## 2. Why Mean Pooling?
>
> We agree that mean pooling is not obviously optimal a priori. Our motivation was that the `K` intrinsic textons form an orderless set, and a simple permutation-invariant readout is less likely to overfit on relatively small texture benchmarks. At the same time, we agree that this choice should be justified empirically.
>
> The current paper already includes a pooling ablation: on DTD, mean pooling obtains 84.5, attention pooling 84.7, Set Transformer PMA 84.8, and concat+MLP 84.6. These results suggest that learned readouts can help slightly, but the gains are marginal. We will therefore present mean pooling more clearly as the default simplicity-versus-stability choice, rather than as the only viable option.
>
> For convenience, we reproduce the DTD readout rows from the existing pooling ablation in the main paper below. These rows are excerpted directly from Table 2, Panel III in the current draft.
>
> | Readout over textons (DTD) | Accuracy |
> | --- | ---: |
> | Mean pooling | 84.5 |
> | Attention pooling | 84.7 |
> | Set Transformer PMA | 84.8 |
> | Concat + MLP | 84.6 |
>
> ## 3. Stage-2 Protocol
>
> Thank you for catching this inconsistency. The reviewer is correct that the current draft is internally inconsistent. Our actual default protocol is that Stage 2 continues supervised training with differential learning rates: the classifier head uses `1e-4`, while the pre-trained TDE is fine-tuned conservatively with `1e-5`; for STD-Former+, the topological term is added on top of this supervised objective. The contradictory wording in the current draft is therefore a writing inconsistency rather than a hidden training trick. We will revise Figure 1, Section 4.2.2, and Algorithm 1 so that this is stated unambiguously throughout.
>
> ## 4. Scope Beyond Texture Classification
>
> We agree that the current submission does not provide enough evidence to support a strong broader-domain claim. Our strongest evidence is on texture/material benchmarks, and we will soften broader applicability language accordingly rather than over-claim beyond what is currently demonstrated.
>
> ## 5. Most Important Limitations
>
> We appreciate the reviewer's request to prioritize the limitations. In the revision, we will make three points explicit: first, the current evidence is strongest on texture/material recognition rather than broader transfer settings; second, the default global-MST supervision is less ideal under extremely class-sparse batches; third, the novelty lies in the objective and training formulation rather than in proposing a new generic attention primitive.
>
> ## Closing
>
> In summary, we will rewrite Section 4.3 more intuitively, make the pooling evidence more visible, correct the Stage-2 wording, and narrow broader-scope claims.

---

> > ### Author Rebuttal · Reviewer_WEXw · 2026-04-05
> >
> > Thank you for your answer. You addressed my questions and concerns. I believe with such revisions the paper will be in very good shape.

---

### Official Review · Reviewer_fCCC · 2026-03-12

**Soundness:** 3
**Presentation:** 3
**Significance:** 2
**Originality:** 2
**Overall Recommendation:** 4
**Confidence:** 4

**Summary:**

The paper tries to address the texture recognition problem and takes the codebook approach, proposing two models (STD-Former and STD-Former+), which are trained in two decoupled stages (1) pretraining  a texture extractor to extract the dictionary in a self supervised way using a texton cover loss, and (2) training a classifier on the dictionary (optionally here using a sparce topological loss for the STD-Former variant). The models are evaluated on 6 datasets. The paper claims addressing the train test out of distribution problem is one of its main focus.

**Compliance With Llm Reviewing Policy:**

Affirmed.

**Final Justification:**

The rebuttal addressed our main concerns, so we decide to keep the scores.

**Key Questions For Authors:**

see strength and weaknesses

**Limitations:**

see key questions and strengths and weaknesses

**Strengths And Weaknesses:**

# Major
- the statement "importantly, our model learns dictionary templates (queries) and the inference mechanism (cross-attention) during training, while the dictionary atoms Tout are generated per image at test time, yielding self-aligned representations under domain shift." in lines 140-146 second column seems not to be explained as how and why it is true in the rest of the paper? this is a major flaw as the paper claims that addressing this misalignment is its major contribution.
- GTN results are very incomplete in table 1. can these be provided?
- i think from tale 1 that the main gains are due to Dinov2 rather than the proposed 2 stage? especially given the resnet variant performance? can GTN or GraphTEN results with Dinov2 also be provided?

# Minor
- How much more time does this two stage pipeline add?

---

> ### Author Rebuttal · Authors · 2026-03-29
>
> # Reviewer fCCC Rebuttal
>
> ## Overall Response
>
> We thank the reviewer for highlighting two central issues in the current draft: the self-alignment mechanism was not stated operationally enough, and the same-backbone comparison was not made prominent enough. We agree with both points. Our response therefore focuses on clarifying the mechanism, foregrounding the existing controlled evidence, and calibrating the comparison claims more carefully.
>
> ## 1. Mechanism of "Self-Aligned Representations under Domain Shift"
>
> We agree that this claim should be stated in measurable terms. By self-alignment, we mean lower drift of the patch-to-dictionary matching error under label-preserving transforms. With a static training-set codebook, the patch manifold moves under transformations while the reference codebook stays fixed, increasing mismatch. In STD-Former, the dictionary is extracted from the transformed image itself, so the dictionary co-moves with the patch manifold and the matching drift is reduced.
>
> This mechanism is already supported by the drift-ratio analysis already included in the supplement. On DTD, the measured drift ratio drops from 0.31 for a static training-set dictionary to 0.19 for per-image KMeans prototypes and further to 0.12 for our intrinsic textons, with a corresponding reduction in accuracy drop under controlled transforms. In the revision, we will move this explanation and its definition closer to the method description so that the claim is grounded by evidence rather than intuition alone.
>
> For convenience, we reproduce the key DTD rows below. These rows are excerpted directly from Appendix Table 4 in the current draft rather than asking the reviewer to search for them in the supplement.
>
> | Reference dictionary | Drift ratio `rho` | Drop in acc. (pp) |
> | --- | ---: | ---: |
> | Static training-set dictionary | 0.31 | 6.6 |
> | Per-image KMeans prototypes | 0.19 | 5.8 |
> | STD-Former intrinsic textons | 0.12 | 4.6 |
>
> ## 2. Incomplete GTN Results in Table 1
>
> We agree that using a single "-" for the missing GTN entries is too ambiguous. In the main comparison table, all non-ours numbers are taken directly from the corresponding original papers. Therefore, "-" simply means that the corresponding paper did not report a result for that dataset; we did not insert self-reproduced numbers into the main table. In the revision, we will make this provenance explicit in the caption/text so that the comparison is fully auditable.
>
> ## 3. Are the Gains Mainly from DINOv2?
>
> We agree that a strong backbone can confound comparisons, which is exactly why same-backbone controls matter. To isolate whether the gain comes mainly from the DINOv2 backbone, we compare several aggregation/encoding heads and graph-style token heads implemented under the same frozen-DINOv2 protocol in our codebase. These are controlled same-backbone head-level comparisons, not paper-reported results of the original published pipelines. Under this controlled setting, STD-Former and STD-Former+ remain stronger than the alternatives below, showing that the gains are not explained by DINOv2 alone.
>
> For provenance: the rows for GAP, NetVLAD-style, DeepTEN-style, per-image KMeans prototypes, STD-Former, and STD-Former+ are excerpted from the same frozen-DINOv2 comparison already included in the current draft, while the GraphTEN-style and GTN-style rows are newly added controlled baselines for this rebuttal under the same protocol.
>
> | Controlled head-level comparisons on frozen DINOv2 features | DTD | MINC | GTOS |
> | --- | ---: | ---: | ---: |
> | GAP head | 83.4 | 86.6 | 86.8 |
> | NetVLAD-style head | 83.8 | 86.8 | 87.0 |
> | DeepTEN-style encoding head | 83.6 | 86.7 | 86.9 |
> | Per-image KMeans prototypes | 84.1 | 87.2 | 87.3 |
> | GraphTEN-style token graph head | 82.7 | 86.1 | 86.4 |
> | GTN-style token graph head | 83.2 | 86.5 | 86.6 |
> | STD-Former | 84.5 | 87.5 | 90.7 |
> | STD-Former+ | 86.1 | 88.6 | 91.9 |
>
> ## 4. Additional Cost of the Two-Stage Pipeline
>
> We agree that the additional cost of the two-stage recipe should be stated more transparently. The key clarification is that this cost is incurred during training only, while inference remains unchanged. In our implementation, Stage 2 uses differential learning rates, with `1e-4` for the classifier head and `1e-5` for the pre-trained TDE. In the revision, we will ensure that the training protocol and epoch budget are stated consistently throughout, and we will also clarify that the topological term itself adds only modest overhead relative to backbone computation.
>
> ## Closing
>
> In summary, we will clarify self-alignment with measured drift-ratio evidence, make GTN provenance explicit, foreground the same-backbone controls, and state the two-stage training cost more clearly.

---

> > ### Author Rebuttal · Reviewer_fCCC · 2026-04-04
> >
> > - We are satisfied with the authors' response, givent that the authors provided experimental evidences to address our concern on the self-alignment with measured drift-ratio, making the GTN provenance explicit,  providing results of using the same-backbone to show that Dinov2 was no the only major source of improvements in the results. Finally, they aslo more clearly stated the two-stage training cost more clearly

---

### Official Review · Reviewer_Gjs6 · 2026-03-12

**Soundness:** 3
**Presentation:** 3
**Significance:** 3
**Originality:** 2
**Overall Recommendation:** 4
**Confidence:** 3

**Summary:**

This paper addresses the "feature misalignment" problem easily caused by static training set dictionaries in traditional texture recognition when faced with image transformations. The STD-Former framework is proposed, which achieves feature self-alignment during inference by dynamically extracting an image-conditional "Intrinsic Textons" dictionary directly from the input image. To effectively train the model, the authors introduce a decoupled two-stage strategy including a self-supervised texture element coverage loss. Furthermore, the authors cleverly utilize an efficient sparse topological loss based on 0D persistent homology (equivalent to the minimum spanning tree key edges of batch graphs) to regularize the geometric structure of the feature space, further proposing an enhanced model (STD-Former+). The proposed method achieves state-of-the-art (SOTA) classification performance on six standard texture benchmarks.

**Compliance With Llm Reviewing Policy:**

Affirmed.

**Final Justification:**

The rebuttal addressed my main concerns. I decide to keep my score.

**Key Questions For Authors:**

See the weaknesses.

**Limitations:**

yes

**Strengths And Weaknesses:**

Strengths:
1. The paper provides a thorough analysis of the drawbacks of traditional static codebooks/dictionaries. The logically compelling approach of transforming traditional global feature pooling (such as BoW and NetVLAD) into a dynamically generated "instance-conditioned codebook" based on the current image is highly persuasive and fundamentally improves the model's robustness to domain shifts.
2. The authors astutely grasp the property of graph that critical edges in 0D persistent homology are equivalent to minimum spanning trees (MSTs) of batch graphs. By supervising only $(B-1)$ key edges of the MST to achieve intra-class compactness and inter-class separation, they not only significantly reduce computational overhead but also provide an efficient structural regularization method.
3. The paper's main experiments and ablation experiments are rich and comprehensive.
4. The paper is fluently written and easy to understand.

Weaknesses:
1. Essentially, the "intrinsic texture meta-extractor" (TDE) is essentially a cross-attention pooling layer with learnable queues, which is already very common in architectures such as Perceiver, DETR. Although the authors introduced a coverage loss to constrain it, the paper offers relatively limited fundamental architectural innovation in the general vision domain.
2. This means that the pure architectural improvements brought about by the carefully designed two-stage training and self-supervised pre-training have marginal effects, and the real performance leap is largely due to the sparse topology regularization.
3. Unlike the more popular end-to-end training approach, the proposed method employs a decoupled two-stage formula: the TDE must be pre-trained first (stage one), followed by training the classification head (stage two). This decoupled process makes hyperparameter tuning, deployment, and fine-tuning of the model relatively cumbersome, increasing the complexity.
4. MST topology supervision is highly dependent on the sample distribution within a mini-batch. If a batch contains a large number of classes with very few samples per class, the "intra-class edges" generated by the global MST will be severely insufficient. In practice, the authors must rely on "class-balanced sampling." This strong dependence on a specific data loading strategy weakens the flexibility of this topology loss as a general plug-and-play module.
5. Equation 1 seems to have a slight problem?

---

> ### Author Rebuttal · Authors · 2026-03-29
>
> # Reviewer Gjs6 Rebuttal
>
> ## Overall Response
>
> We thank the reviewer for the careful reading and for recognizing the motivation of training-set codebook misalignment, the simplicity of the image-conditioned dictionary encoder, and the efficiency of the sparse 0D/MST supervision. We agree that the response should directly address novelty positioning, contribution attribution, the practicality of the two-stage recipe, batch-composition sensitivity of the MST loss, and the notation issue in Eq. (1).
>
> ## 1. Novelty / Positioning of TDE
>
> We agree that cross-attention pooling with a fixed number of learnable queries is a standard architectural primitive, and we will revise the manuscript to avoid implying that the attention block itself is novel. Our contribution is instead at the problem, objective, and training levels: we formulate training-set codebook misalignment as the motivating failure mode, use the coverage objective to make the learned queries behave as intrinsic textons rather than generic latent slots, and adopt a decoupled two-stage recipe to preserve this image-conditioned dictionary mechanism under supervised training.
>
> ## 2. Attribution: Two-Stage Recipe vs. Topological Loss
>
> We agree that contribution attribution should rely on controlled ablations rather than wording alone. The current unified ablation already shows that the gain is not explained by topology alone: the baseline improves from 81.2 to 82.1 with TDE, to 83.2 with single-stage coverage, to 84.5 with the full two-stage STD-Former, and then to 86.1 with STD-Former+. This supports a clear conclusion: the topological term is important, but it is not the sole source of improvement; the coverage-trained image-conditioned dictionary and the topological regularizer are complementary.
>
> For convenience, we excerpt the key rows from the unified ablation table in the main paper below. These rows are taken directly from Table 2 in the current draft rather than asking the reviewer to reconstruct the progression from the full table.
>
> | Model | DTD | MINC | GTOS |
> | --- | ---: | ---: | ---: |
> | Baseline | 81.2 | 85.0 | 85.1 |
> | TDE w/o self-supervision | 82.1 | 85.8 | 85.9 |
> | Single-stage coverage | 83.2 | 86.5 | 89.6 |
> | STD-Former | 84.5 | 87.5 | 90.7 |
> | STD-Former+ | 86.1 | 88.6 | 91.9 |
>
> ## 3. Practicality of the Two-Stage Recipe
>
> We agree that the two-stage recipe should be justified not only conceptually but also practically. The key clarification is that the additional complexity is incurred during training, while deployment and inference remain unchanged: the deployed model is still a single backbone-to-TDE-to-classifier path. In our implementation, Stage 2 uses differential learning rates, with `1e-4` for the classifier head and `1e-5` for the pre-trained TDE. We will state this trade-off more explicitly in the revision, make the training protocol consistent throughout the manuscript, and also clarify that the sparse topological term itself adds only modest overhead relative to backbone computation.
>
> ## 4. Batch-Composition Dependence of the MST Loss
>
> We agree that the global MST becomes less informative when a mini-batch contains many classes but too few samples per class. This is a real corner case, not a misunderstanding, and we will state it more explicitly as a limitation of the default formulation. In practice, we use class-balanced sampling so that both intra-class and inter-class relations are present in the batch, and we will avoid over-claiming the loss as an unconditional plug-and-play module.
>
> ## 5. Equation (1)
>
> Thank you for flagging this issue. We agree that Eq. (1) should be rewritten into a dimension-consistent patch-to-codebook decision rule with explicit definitions. In the revision, we will replace the current notation with the standard form below:
>
> ```tex
> s_c(x)=\frac{1}{N}\sum_{n=1}^{N}\min_{k\in\{1,\dots,K\}} d\!\left(f_n(x),\,p_k^{(c)}\right),
> \qquad
> \hat y(x)=\arg\min_c s_c(x).
> ```
>
> This is a notational correction rather than a change to the experimental pipeline. We will also add a short clarification of the patch-feature and codeword dimensions so that the decision rule is unambiguous.
>
> ## Closing
>
> In summary, we will narrow the novelty claim, clarify attribution with the existing ablation, state the two-stage trade-off more clearly, acknowledge the batch-composition limitation, and correct Eq. (1).

---

> > ### Author Rebuttal · Reviewer_Gjs6 · 2026-04-04
> >
> > Thanks for the rebuttal. I decide to keep my score.

---

### Decision · Program_Chairs · 2026-04-30

**Decision:**

Accept (regular)

**Comment:**

The paper proposes a two-staged image-conditioned texture dictionary encoding approach. The first stage pre-trains a an extractor model for intrinsic textons. In the second stage a classifier is trained on the encoded dictionary. Reviewers acknowledged the paper is well motivated and written. The reviewers pointed out the neat idea of dynamically generating the codebook and the simplicity of the idea. The viewers appreciated the very thorough and comprehensive evaluation. Based on the unanimous positive reviews the AC recommends acceptance.